# Immune aging impairs muscle regeneration via macrophage-derived anti-oxidant selenoprotein P

Dieu-Huong Hoang [1,4], Jessica Bouvière [1,4], Johanna Galvis[1,4], Pauline Moullé [1], Orane Mercier [1], Eugenia Migliavacca [2], Ananga Ghosh[1], Gaëtan Juban [1], Sophie Liot[1], Pascal Stuelsatz [2], Fabien Le Grand[1], Jérôme N Feige [2,3,5], Rémi Mounier [1,5✉] & Bénédicte Chazaud [1,5✉]

## Abstract

**Muscle regeneration is impaired with aging, due to both intrinsic defects of muscle stem cells (MuSCs) and alterations of their niche. Here, we monitor the cells constituting the MuSC niche over time in young and old regenerating mouse muscle. Aging alters the expansion of all niche cells, with prominent phenotypes in macrophages that show impaired resolution of inflammation. RNA sequencing of FACS-isolated mononucleated cells uncovers specific profiles and kinetics of genes and molecular pathways in old versus young muscle cells, indicating that each cell type responds to aging in a specific manner. Moreover, we show that macrophages have an altered expression of Selenoprotein P (Sepp1). Macrophage-specific deletion of *Sepp1* is sufficient to impair the acquisition of their restorative profile and causes inefficient skeletal muscle regeneration. When transplanted in aged mice, bone marrow from young WT mice, but not Sepp1-KOs, restores muscle regeneration. This work provides a unique resource to study MuSC niche aging, reveals that niche cell aging is asynchronous and establishes the antioxidant Selenoprotein P as a driver of age-related decline of muscle regeneration.**

**Keywords** Aging; Macrophages; Skeletal Muscle Regeneration; Selenoprotein P
**Subject Categories** Immunology; Musculoskeletal System; Stem Cells & Regenerative Medicine

## Introduction

Skeletal muscle is an important determinant in healthy aging, through both adaptative, metabolic and regenerative capacities of muscles that enable sustained contraction and physical performance. Adult skeletal muscle is a plastic tissue and can regenerate after trauma- or exercise-induced myofiber damage via muscle stem cells (MuSCs), that exit quiescence, expand, differentiate and eventually fuse to form new functional myofibers (Sousa-Victor et al, 2022). Although MuSCs are absolutely required for skeletal muscle regeneration, their surrounding non-myogenic counterparts in the local niche coordinate inflammatory signals and tissue remodeling to sustain adult myogenesis (Panci and Chazaud, 2021; Sousa-Victor et al, 2022). However, this process is altered in a variety of conditions, including muscle diseases, some metabolic conditions such as diabetes, and aging. Failure of mounting an efficient skeletal muscle regeneration in aged organisms has been attributed to both intrinsic alterations of MuSCs and modified environmental cues (Hong et al, 2022). Since they are the support of muscle regeneration, a variety of intrinsic alterations have been identified in the old MuSCs, including changes in epigenetics and signaling, as well as alterations in metabolism and proteostasis (reviews in (Hong et al, 2022; Sousa-Victor et al, 2022)). Remarkably, extrinsic alterations have also been described including alterations in the number or in the nature of immune cells (Cui et al, 2019; Kuswanto et al, 2016; Markworth et al, 2021; Paliwal et al, 2012; Patsalos et al, 2018; Rahman et al, 2020; Reidy et al, 2019; Sloboda et al, 2018; Zhang et al, 2020), in some properties of fibro-adipogenic precursors (FAPs) (Lukjanenko et al, 2019; Zwetsloot et al, 2012) and in extracellular matrix (ECM) composition (Cui et al, 2019; Kanazawa et al, 2022; Lukjanenko et al, 2019; Rahman et al, 2020; Schüler et al, 2021; Stearns-Reider et al, 2017), as well as systemic factors (Hong et al, 2022). However, if cell–cell interactions are well-described in the adult regenerating muscle (Panci and Chazaud, 2021; Singh and Chazaud, 2021), the impact of aging on the molecular regulation of cell components of the MuSC niche and on cell–cell interactions during regeneration is still poorly known.

Here, we compared and analyzed the time course of the various cell types constituting the MuSC niche during muscle generation in young and old mice. We showed that all cells showed alterations in their kinetics and particularly macrophages, which exhibited an impaired resolution of inflammation in the old regenerating muscle. From RNA sequencing of FACS-isolated MuSC niche cells before and 2, 4, and 7 days after the muscle injury, we extracted point-by-point and longitudinal analyses that define cell-specific

[1]Institut NeuroMyoGène, Unité Physiopathologie et Génétique du Neurone et du Muscle, Université Claude Bernard Lyon 1, Inserm U1315, CNRS 5261, Lyon, France. [2]Nestlé Institute of Health Sciences, Nestlé Research, Lausanne, Switzerland. [3]School of Life Sciences, Ecole Polytechnique Fédérale de Lausanne (EPFL), Lausanne, Switzerland. [4]These authors contributed equally: Dieu-Huong Hoang, Jessica Bouvière, Johanna Galvis. [5]These authors contributed equally: Jérôme N Feige, Rémi Mounier, Bénédicte Chazaud. ✉E-mail: remi.mounier@univ-lyon1.fr; benedicte.chazaud@inserm.fr

signatures of aging and regeneration in the muscle stem cell niche. These results, that are made publicly available via an online resource, indicate that aging is asynchronous in the MuSC niche, with each cell type responding to aging and impacting tissue repair in a cell- and time-specific manner. Finally, we discovered a new role for Selenoprotein P (Sepp1) that was downregulated in old repairing macrophages. Macrophage-specific deletion of *Sepp1* gene impaired the resolution of inflammation, altered the interactions between macrophages and MuSCs, and impacted the efficiency of skeletal muscle regeneration.

# Results

## Regeneration is impaired in old skeletal muscle

Adult (3 months, hereafter called young) and old (24 months) male mice were injected with cardiotoxin in the *Tibialis Anterior* muscle and the muscles were collected 1, 2, 4, 7, or 28 days after the injury for histological and flow cytometry analyses. The muscle mass/body weight was slightly different at steady state, old muscles being 6.1% lighter than young muscles. 28 days after the injury, old muscle showed a reduction in weight of 21.3% as compared with the young ones (Fig. 1A), while old mice showed a slight increase in their body weight (Fig. EV1A). The cross-sectional area of the regenerating myofibers was decreased at day 7 (−14%) and remained lower at day 28 (−26.3%) post injury in old mice where notably the number of large myofibers was strongly reduced (Figs. 1B,C and EV1B,C). Accordingly, the overall area of the muscles was lower after regeneration in the aged animals (−17.3%, Fig. EV1D), while the number of myofibers was increased (+28.9%) (Fig. 1D). Thus, a decreased muscle mass, together with smaller and more numerous regenerating myofibers are indicative of an impairment of the regeneration process in the old muscle.

The necrosis following muscle damage was identical in young and old muscles following injury, but was longer to resorb in the old muscles, indicative of an age-related defect in the cleansing of muscle debris (Figs. 1E and EV1E). However, the impaired regeneration was not associated with an increase in the collagen area in the old animals (Figs. 1F and EV1F). After a transient increase at day 7 post injury in the old compared with young muscle, the number of fibro-adipogenic precursors (FAPs) was similar in both muscles at the end of the regeneration process (Figs. 1G and EV1G). Vascular remodeling was also affected by age only at day 7 post injury (−27.4%) and was back to the values observed in the young muscle at day 28 (Figs. 1H and EV1H). Macrophage density was also altered in the old regenerating muscle. At day 7 after injury, a time point when the resolution of inflammation is operated in normal adult regenerating muscle (Varga et al, 2016a), the number of macrophages was notably elevated in the old muscle (+144.5%) (Figs. 1I and EV1I), suggesting a failure in the resolution of inflammation. Thus, the kinetics of the pro-inflammatory (CD64$^{pos}$Ly6C$^{pos}$) and restorative (CD64$^{pos}$Ly6C$^{neg}$) macrophage populations were then analyzed at days 2, 4, and 7 after the injury by flow cytometry. During the time course of the resolution of inflammation, i.e. from day 2 to day 4 post injury (Varga et al, 2016a), the number of Ly6C$^{pos}$ pro-inflammatory macrophages was +41% and +92% higher in the old muscle, respectively (Fig. 1J). At day 7, the number of Ly6C$^{pos}$

inflammatory macrophages was still 60% higher in the old regenerating muscle than in the young one (Fig. 1J) to the detriment of Ly6C$^{neg}$ restorative macrophages that were less numerous in the old regenerating muscle. Altogether, these results demonstrate that aging impairs the resolution of inflammation in macrophages and alters the acquisition of the restorative phenotype required to support efficient myofiber regeneration (Arnold et al, 2007; Saclier et al, 2013).

## Kinetics of gene expression in the various cell types controlling skeletal muscle regeneration reveal asynchronous aging

To identify the kinetics of gene expression in the various cell types involved in skeletal muscle regeneration, MuSCs, Endothelial Cells (ECs), FAPs, Ly6C$^{pos}$ (inflammatory) macrophages, Ly6C$^{neg}$ (resolving) macrophages and neutrophils were FACs-isolated from young and old muscle before and at days 2, 4, and 7 after the injury (Appendix Fig. S1) using the gating strategy reported in Juban et al 2018. Bulk RNAseq analysis was performed on all the conditions whenever it was possible to sort cells, using a low-input library preparation kit and paired-end sequencing.

The correlation analysis shows that immune cells gathered while FAPs and MuSCs were correlated (Fig. 2A). As expected, the effect of the cell type was the strongest contributor to differences in gene expression in the PCA analysis (Fig. 2B). All three myeloid cell types (neutrophils, inflammatory and resolving macrophages) clustered together while MuSCs, FAPs and ECs clustered as discrete populations (Fig. 2B). Among cell populations, the second level of correlation was the time point after injury. Indeed, the gene signature of each cell type differed according to the time after injury (Fig. 2C). Finally, the last correlation analysis was the age and all conditions (cell type and time point) except 2 (ECs and inflammatory macrophages at day 2), showed the segregation between cells isolated from old and young muscles (Fig. 2D).

We then analyzed the differential enrichment in the molecular pathways for all conditions in old versus young conditions (Fig. 2E). The size of the circles (and size of lettering) was correlated with the number of pathways that were enriched and the coloring correlated with the number of conditions (cell type and time point) in which those pathways were differentially enriched. The pathways the most impacted by age in several cell types and that affected numerous conditions were cell cycle, metabolism, signal transduction, transcription and DNA repair, as well as extracellular matrix organization. The reader can refer to the online report to zoom in on the various pathways or to extract the enriched pathways from the 36 conditions encompassing the 6 cell types, 4 time points and 2 ages (https://github.com/LeGrand-Lab/Ageing-impact_in_gene_expression_on_skeletal_muscle_repair).

When looking by cell type and day post injury (Fig. 2F), a general tendency was that most pathways were downregulated in resting (day 0) and early regenerating (day 2) old muscle, while molecular pathways were upregulated during the later stages of muscle regeneration (days 4 and 7). At day 0 and day 2 after the injury, we observed that most of the differentially expressed pathways were downregulated, while they were upregulated at day 4. At day 7, only MuSCs showed an enrichment in downregulated pathways (Fig. 2F). However, there were some differences in how the various cell types responded to muscle injury in the old muscle. MuSCs were the cell type in which age affected the most the

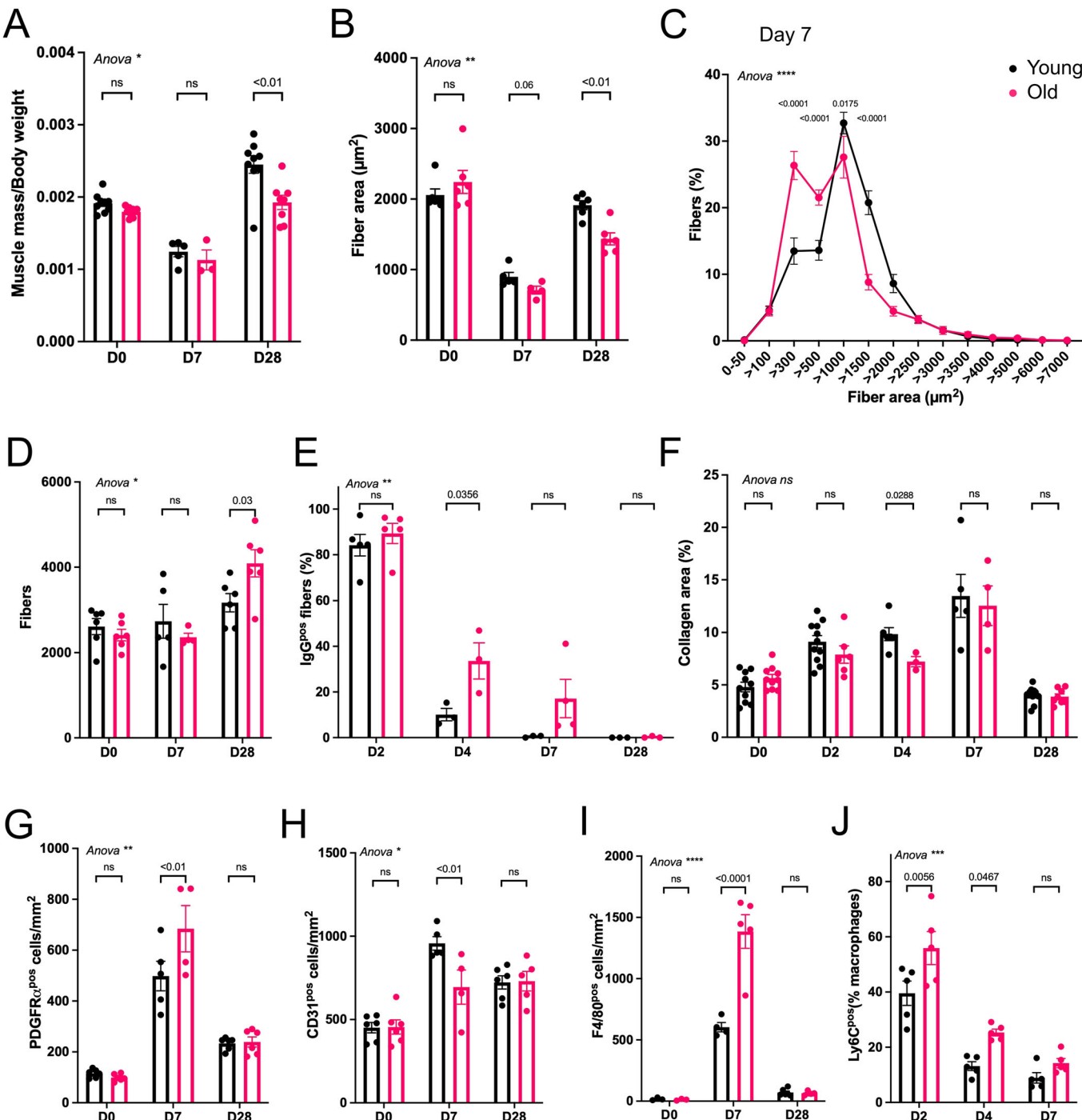

activation/repression of molecular pathways, all along the regeneration process. ECs from old muscle showed a specific response in pathway enrichment, mainly by upregulation of pathways during the restorative phase of muscle regeneration. Inversely, old FAPs mainly exhibited downregulation of pathways, compared to young cells, and were observed during the first days after the injury. These results exemplify the complexity of aging at the cellular level, where each individual cell type may present specific aging trajectories (Rutledge et al, 2022).

However, when zooming in each condition (Fig. EV2), we observed that in ECs, FAPs and MuSCs, an enrichment in the immune system-related pathways was found upregulated indicating that all these non-immune cell types increased their expression of inflammatory cues in the old regenerating muscle. Another commonality between ECs, FAPs and MuSCs isolated from old regenerating muscle is that they downregulated the expression of genes associated with ECM organization (Fig. EV2). Inflammatory macrophages also increased their inflammatory profile, while both

Figure 1. **Histological analysis of regenerating young and old muscle.**

*Tibialis Anterior* muscles from young (10 weeks old) and old (24 months old) mice were injected or not with cardiotoxin and were harvested 2, 4, 7, and 28 days after the injury. (A) The muscle mass (mg) was normalized to the body weight (mg) ($n = 3$–9). A two-way ANOVA test was performed, followed by multiple comparisons using Šidák test. (B–I) The muscle sections were immunostained for various proteins. (B–D) From laminin immunostaining, the mean cross-section myofiber area (B), cross-section myofiber area distribution at day 7 after injury (C), and the total number of myofibers per muscle section (D) were measured ($n = 4$–6). A two-way ANOVA test was performed, followed by multiple comparisons using Šidák test. (E) From IgG immunostaining, the proportion of positive myofibers, indicative of necrotic myofibers, was quantified as a percent of total myofibers ($n = 3$–5). A two-way ANOVA test was performed, followed by multiple comparisons using Šidák test. (F) From Collagen I immunostaining, the fibrosis area was quantified as a percentage of the total field. ($n = 3$–11). The two-way ANOVA test was nonsignificant. Multiple unpaired $t$ tests were performed, and the $P$ values are given for each day. (G) From PDGFRα immunostaining, Fibro-Adipogenoc Progenitor (FAP) number was quantified ($n = 4$–6). A two-way ANOVA test was performed, followed by multiple comparisons using Šidák test. (H) From CD31 immunostaining, the number of endothelial cells was quantified ($n = 4$–6). A two-way ANOVA test was performed, followed by multiple comparisons using Šidák test. (I) From F4/80 immunostaining, macrophage numbers were quantified ($n = 3$–6). A two-way ANOVA test was performed, followed by multiple comparisons using Šidák test. (J) The number of Ly6C[pos] inflammatory macrophages was quantified by flow cytometry as a percentage of CD45[pos] immune cells ($n = 5$). A two-way ANOVA test was performed, followed by multiple comparisons using Šidák test. Data information: Values are given as mean ± SEM. Each dot represents one mouse. The result of the two-way ANOVA test is shown for each graph as *$P < 0.05$; **$P < 0.01$; ***$P < 0.001$; ****$P < 0.0001$. Source data are available online for this figure.

macrophage populations showed an increased expression of the cell cycle pathway (Fig. EV2).

Next, we analyzed the differentially expressed genes (DEGs) in the various conditions and found different kinetics according to the cell type considered (Fig. 3A,B). ECs showed an increased amount of DEGs only at D4 and D7 after the injury while FAPs and resolving macrophages exhibited a continuous differential expression of genes at all time points, including at steady state. MuSCs showed strong differential gene expression at D2 and D4 after injury (Fig. 3A). Details on DEGs are provided in volcano plots in Fig. EV3 and in the interactive DEG report (https://github.com/LeGrand-Lab/Ageing-impact_in_gene_expression_on_skeletal_muscle_repair). Longitudinal kinetics analysis allowed the identification of DEGs at various time points along the regeneration process that are represented by the colored lines under the loops (Fig. 3B). For instance only a few genes were consistently differentially expressed between old and young in MuSCs and FAPs at the four time points (Fig. 3B, brown line) while a high number of DEGs were present at D2 and D4 after the injury in MuSCs (Fig. 3B, regular blue line). Details on DEGs for each condition and combination of conditions are available in the interactive DEG report, which further allows seeking a specific gene. Zooming in on the DEG analysis, we separated upregulated and downregulated DEG for each cell type/time point (Appendix Fig. S2). For all conditions, similar numbers of DEG were found to be upregulated and downregulated (Appendix Fig. S2). However, the kinetics were different according to the cell type. In ECs, FAPs and inflammatory macrophages, almost all DEGs followed the same kinetics during the time course of muscle regeneration, being either up or downregulated along the process (Appendix Fig. S2). On the contrary, in MuSCs, numerous DEGs were first downregulated (until D2 or D4) and then upregulated (from D2 or D4) in the old muscle (light purple lines, Appendix Fig. S2). These results show how various cell types asynchronously respond to tissue damage in the old muscle, emphasizing the high complexity of aging at the cellular level within the same tissue.

## Kinetics of gene expression in macrophages in old versus young regenerating muscles

Given that the resolution of inflammation was impaired in the old regenerating muscle, we zoomed in the analysis of resolving macrophages. The analytic report provided five genes that were differentially expressed at all three time points of the regeneration in old versus young resolving macrophages. Among those genes,

only one gene showed a unique kinetics, represented by the purple line in Fig. 3C, with upregulation at D2 followed by downregulation during the repair phase of muscle regeneration (D4 and D7). Old resolving macrophages expressed a lower amount of *Sepp1* (*selenop*) transcripts (encoding for Selenoprotein P) at days 4 and 7 after injury as compared to young resolving macrophages ($-27.1\%$ and $-23.4\%$, respectively) (Fig. 3D). Strikingly, the huge increase of *Sepp1* expression that was observed in the young macrophages between day 2 and day 4 after injury, i.e., at the time of the resolution of inflammation ($+320\%$) was twice lower in the old macrophages ($+100\%$) (Fig. 3D), a kinetics evoking a defect in the acquisition of the restorative phenotype. Sepp1 is a secreted glycoprotein belonging to the selenoprotein family (Burk and Hill, 2009; Burk et al, 2003). It possesses two different functions: Selenium (Se) transport activity to supply Se to cells and antioxidant via GPX (glutathione peroxidase)-like activity to reduce phospholipid hydroperoxide (Saito et al, 2004). Using total KO and specific mutated forms of one or the other domain, we investigated the role of Sepp1 in macrophage functions during muscle regeneration in vitro and in vivo.

## Selenoprotein P is required in macrophages for the resolution of inflammation in vitro

Bone marrow-derived macrophages (BMDMs) from Sepp1[KO] mice (Hill et al, 2003) were treated with either IFNγ or IL10 to induce their activation into pro-inflammatory and anti-inflammatory macrophages, respectively (Mounier et al, 2013; Saclier et al, 2013), and were analyzed for their expression of several inflammatory markers by immunofluorescence. Such analysis at the protein level reflects the acquisition of the inflammatory (pro or anti) phenotype (Mounier et al, 2013) (Fig. 4A). As expected in WT macrophages, the expression of the pro-inflammatory markers iNOS and CCL3 was reduced in IL10- *versus* IFNγ-treated macrophages ($-20.3\%$ and $-31.6\%$, respectively) (Fig. 4B,C). This was not observed in Sepp1[KO] macrophages (Fig. 4B,C). Similarly, the increase in the expression of the anti-inflammatory markers CD206 and CD163 observed in IL10-treated WT macrophages ($+25\%$ and $+16\%$ respectively, when compared with pro-inflammatory macrophages) was not observed in Sepp1[KO] macrophages (Fig. 4D,E). This indicates that Sepp1[KO] macrophages did not acquire the anti-inflammatory phenotype upon adequate cytokine stimulation. To assess macrophage function, conditioned

**A** Distance matrix Spearman correlation

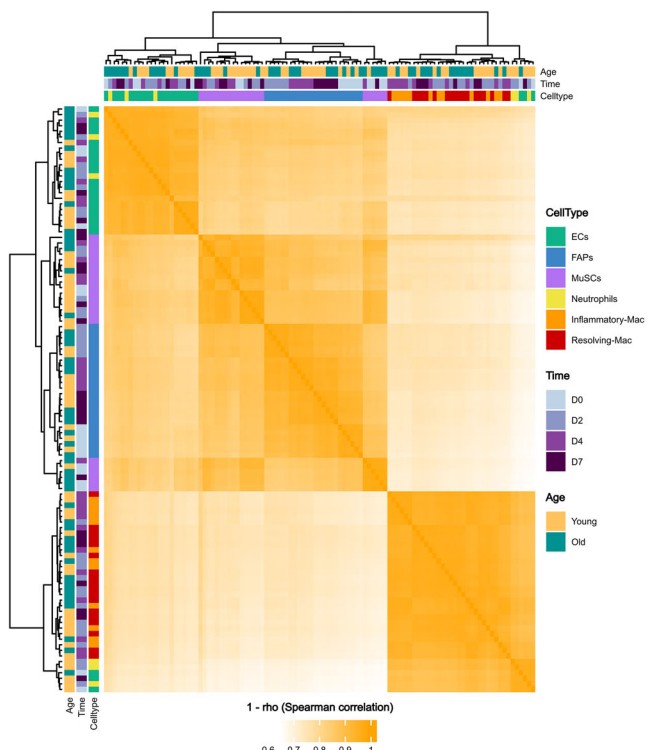

**B** Two first PCA components on whole dataset

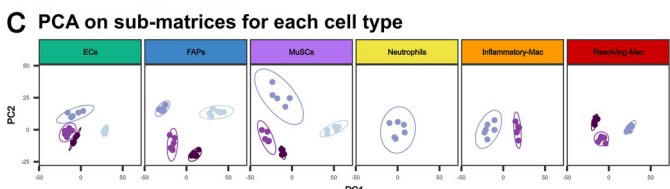

**C** PCA on sub-matrices for each cell type

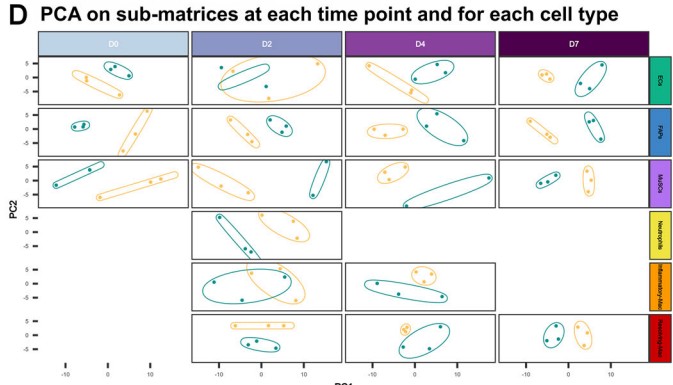

**D** PCA on sub-matrices at each time point and for each cell type

**E** Significant enriched pathways overview Old vs. young

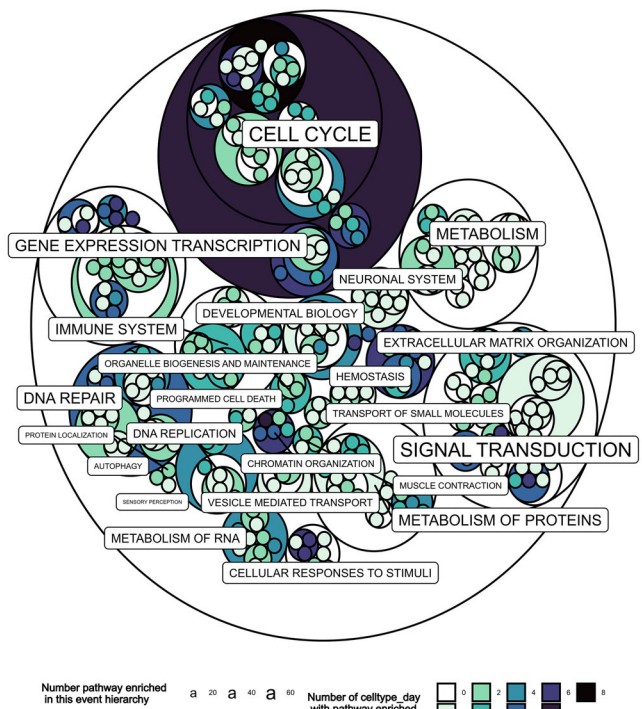

**F** Distribution of signicant enriched pathways in Old vs. young according to log2FoldChange

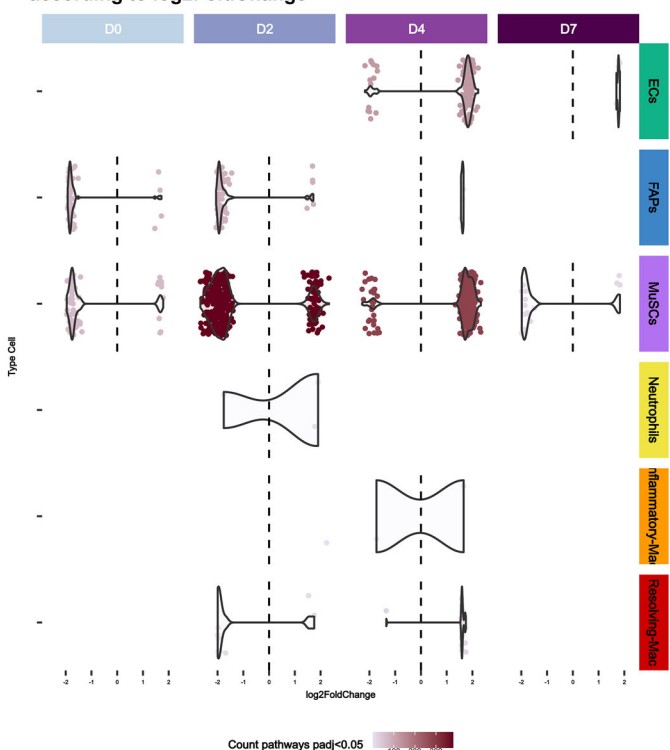

◄ Figure 2. Principal component analysis and enriched signaling pathways in old *versus* young mononucleated cells.

(A) Heatmap of Spearman's correlation coefficients for individual sample replicates isolated from age, time post injury and cell type. Correlation was computed on normalized counts after the preliminary filter. (B) Principal component analysis (PCA) of all 105 replicates based on vst. Principal component (PC) 1 splits the samples in immune cells of other cells and component 2 splits Endothelial cells (ECs), Fibro-Adipogenic Progenitors (FAPs), and Muscle Stem Cells (MuSCs). (C) For each cell type, PCA was done on their replicates. PC1 and 2 split samples by time post injury. (D) For each cell type and time post injury, PCA was done on these replicates: PC1 and 2 split samples by age. (E, F) Presentation of significantly ($Padj <= 0.05$) enriched Reactome pathways (with the GSEA method) with age. (E) A hierarchical overview of Reactome pathways is presented, each label corresponds to one of the 25 top-level pathways, and the label size is scaled based on the number of pathways contained in their pathways' sons. Each circle corresponds to a pathway, and its color represents the number of celltype_day where this pathway was enriched. (F) Violin plots explore the number of enriched pathways (colored points) and their log2 fold change in x axis for each cell type, day_post_injury young vs old samples.

medium from activated BMDMs was used on muscle stem cell (MuSC) culture (Fig. 4F) since we previously showed that pro-inflammatory macrophages activate MuSC proliferation, while anti-inflammatory macrophages activate their differentiation and fusion into myotubes (Arnold et al, 2007; Mounier et al, 2013; Saclier et al, 2013). As expected, IL10-treated WT BMDMs decreased MuSC proliferation ($-25.2\%$ when compared with IFNγ-treated macrophages, Fig. 4G) and increased their fusion ($+77.8\%$ when compared with IFNγ-treated macrophages, Fig. 4H). On the contrary, IL10-treated Sepp1$^{KO}$ BMDMs did exhibit similar functional properties to IFNγ-treated BMDMs (Fig. 4G,H), indicating they did not acquire the anti-inflammatory phenotype. These results indicate that Sepp1 is required for the acquisition of the anti-inflammatory macrophage phenotype and function.

Sepp1 is a secreted glycoprotein that has two functions: it supplies Se to cells via its C-terminus domain which contains 9 selenocysteins, and acts as an antioxidant, via its N-terminal domain that contains one selenocystein in a redox motif (Saito et al, 2004; Saito and Takahashi, 2002). We used two mutants to establish whether one or the other function was necessary to the acquisition of the recovery phenotype by macrophages. The Sepp1$^{U40S/U40S}$ mutant bears a serine instead of the selenocystein involved in the antioxidant activity of Sepp1 (Kurokawa et al, 2014) (Fig. EV4A). The Sepp1$^{\Delta240-361}$ is truncated for the C-terminal domain and is deficient for the selenium transport function of the protein (Hill et al, 2007) (Fig. EV4A). BDMDs from both genotypes gave results similar to those obtained with the total Sepp1 KO. Indeed, they showed a deficiency in the acquiring the anti-inflammatory phenotype upon activation with IL10 (Fig. EV4B–E), and they did not acquire the restorative function towards MuSC myogenesis (Fig. EV4F–I). These results show that both Sepp1 antioxidant and Se transport functions are required in macrophages for the resolution of inflammation and their acquisition of the resolving phenotype and functions.

## Selenoprotein P is required in macrophages for the resolution of inflammation in vivo

We then used the LysM$^{Cre}$;Sepp1$^{fl/fl}$ mouse (hereafter Sepp1$^{\Delta Mac}$) (Hill et al, 2012) to analyze the impact of *Sepp1* deletion in the myeloid lineage on skeletal muscle regeneration in vivo (Fig. 5A). *Sepp1* deletion efficacy was checked in BMDMs (Fig. EV5A). Although LysozymeM is expressed by both neutrophils and macrophages, previous studies have shown that the LysM$^{Cre}$ model is appropriate to specifically investigate macrophage function in skeletal muscle regeneration (Mounier et al, 2013; Varga et al, 2016b). Moreover, there was no impact of *Sepp1* deletion on neutrophil infiltration and kinetics in the regenerating muscle

(Fig. EV5B). Flow cytometry analysis allows to discriminate the sequential steps of macrophage shift from Ly6C$^{pos}$ cells (pro-inflammatory macrophages) to Ly6C$^{neg}$ cells (recovery macrophages), passing by Ly6C$^{int}$ macrophages that are en route to the inflammatory shift (Fig. EV5C). At day 1 after the injury, there was no difference in the distribution of the macrophage subsets, Ly6C$^{pos}$ inflammatory macrophages being the most abundant (Fig. EV5D). Two days after the injury, the number of inflammatory Ly6C$^{pos}$ macrophages was higher to the detriment of Ly6C$^{neg}$ cells in Sepp1$^{\Delta Mac}$ muscle (Fig. EV5E). At day 3 after the injury, Ly6C$^{pos}$ macrophages number increased by 107% while the number of Ly6C$^{neg}$ was lowered by $-14.6\%$ (Fig. 5B). This phenotype was observed until day 4 ($+59\%$ of Ly6C$^{pos}$ macrophages and $-6.5\%$ of Ly6C$^{neg}$ macrophages in Sepp1$^{\Delta Mac}$ muscle as compared with the WT, Fig. EV5F), a time point at which the shift of macrophages is ended in this model (Varga et al, 2016a). Of note, no impact of Cre expression was observed on macrophage populations at day 3 after muscle injury (Fig. EV5G). These results indicate a failure in the acquisition of the resolving macrophage phenotype in Sepp1$^{\Delta Mac}$ muscle.

The consequence of the failed resolution of inflammation on muscle regeneration was a strong increase in the number of regenerating myofibers expressing the embryonic isoform of the Myosin Heavy Chains 7 days after the injury ($+34.3\%$ in Sepp1$^{\Delta Mac}$ muscle vs. WT) (Fig. 5C), indicating a delayed formation and maturation of the new myofibers. Ultimately, the number of myonuclei per myofiber (similar in the two genotypes in the uninjured muscle, Fig. EV5H) one month after the injury was found 30.2% smaller in Sepp1$^{\Delta Mac}$ than in WT animals (Fig. 5D). This result is in accordance with the defect of myogenesis described above in vitro in the presence of Sepp1$^{\Delta Mac}$ macrophages. However, these defects only modestly impacted the size of myofibers one month after the injury (Fig. EV5I), suggesting a transient impact of macrophage Sepp1 deficiency on the overall regeneration process. These data show an alteration of the regenerative capacities of the muscle in Sepp1$^{\Delta Mac}$ mice, demonstrating that macrophagic Sepp1 is involved in the resolution of inflammation. A similar extent in alteration of macrophage shift was observed in other genotypes (Mounier et al, 2013; Saclier et al, 2020; Tonkin et al, 2015), linking macrophage shift dynamics to the efficacy of muscle regeneration (Varga et al, 2016a).

To further link Sepp1 deficiency in macrophages with aging, we performed bone marrow transplantation experiments in which old (24 m.o.) mice were irradiated and transplanted with bone marrow from either young WT, old WT, or young Sepp1$^{\Delta Mac}$ mice (Fig. 5E). One month later, muscle was injured and analyzed for regeneration at days 7 and 28. It is to be mentioned that irradiation strongly delays muscle regeneration in adult mice, impacting notably the

**A** Distribution of significantly differentially expressed genes (DEGs) Old vs Young, according to log2FoldChange

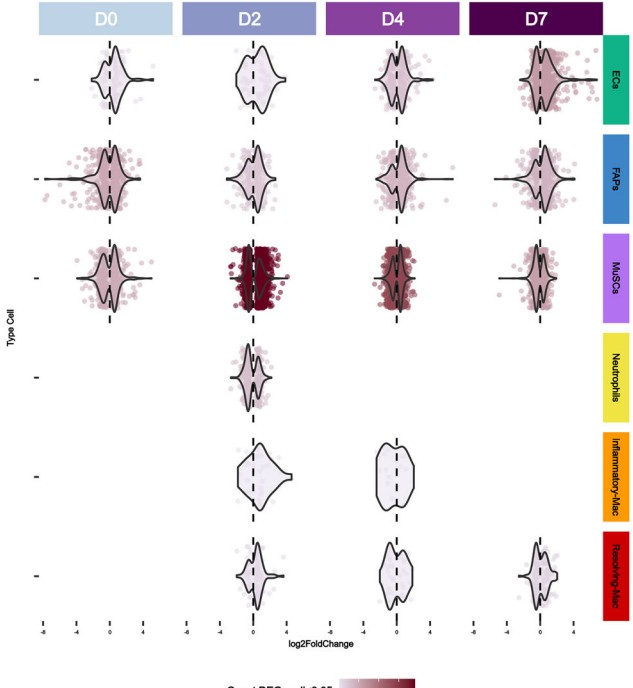

**B** Overview of DEGs at day and during consecutive days in each cell type

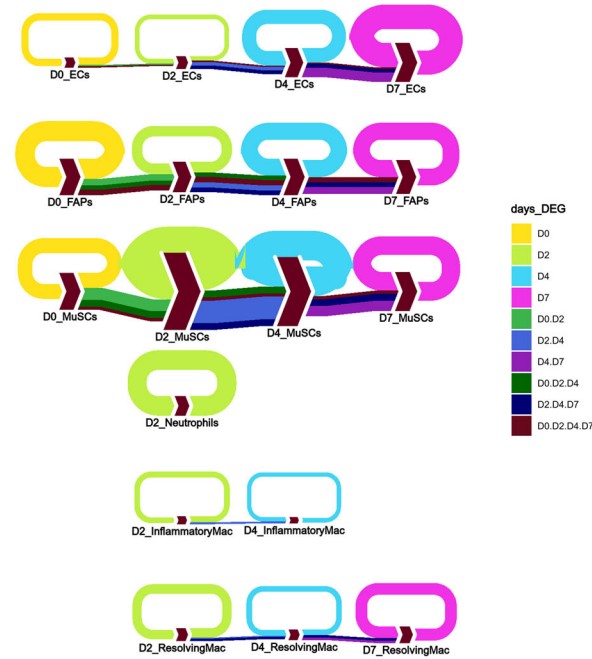

**C** Overview of UP and DOWN DEGs in resolving macrophages

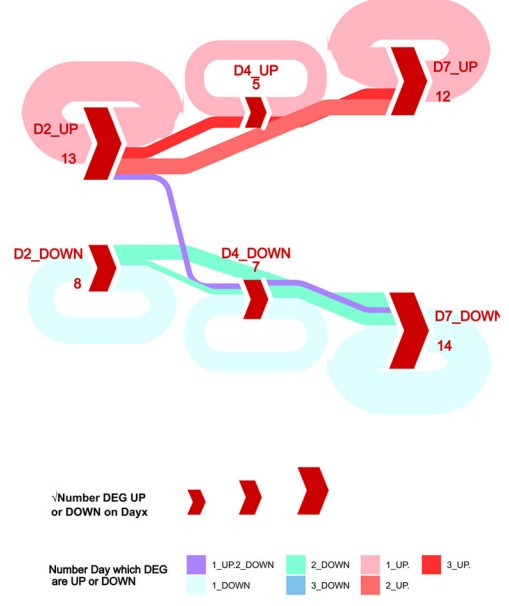

**D** Selenoprotein expression in old and young resolving macrophages

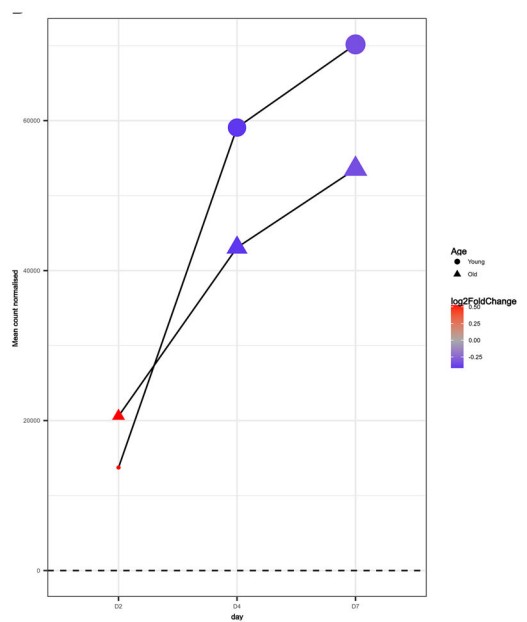

MuSC pool (Patsalos et al, 2017; Varga et al, 2016b), and we found this phenomenon exacerbated in the old animals. For instance, the number of eMHC[pos] myofibers was much lower in that condition than in non-irradiated mice (Fig. EV5J, to be compared with Fig. 5C WT [Sepp1 + ]), indicative of a slower process of regeneration. We observed that, as compared with old mice transplanted with young bone marrow, the area of regenerating eMHC[pos] myofibers was decreased in mice transplanted with old bone marrow, and this was not rescued by young bone marrow deficient for Sepp1 (Fig. 5F). At that time point, neither the number

Figure 3. Differentially expressed genes (DEG) in old *versus* young mononucleated cells.

(A) Violin plots explore the number of DEGs, each color point is a DEG (i.e., 1905 DEG in D2 MuSCs) and their log2foldchange in *x* axis for each cell type, day_post_injury young *vs*. Old samples. (B) DEG proportion at one day (loops) or on several consecutive days (lines) to analyze if ageing impacts on gene expression in one cell type specifically at one or several time points during muscle regeneration. The thickness of the loops and the lines correlate with the number of DEGs. (C) DEG cycle and flow in resolving macrophages during regeneration with segregation of upregulated genes (reddish colors) and downregulated genes (blueish colors). Note the purple flow showing one gene upregulated at D2 then downregulated at D4 and D8. (D) Zoom expression of Sepp1 transcript in resolving macrophages at D2, 4, and 8 after injury.

of MuSCs nor that of FAPs were altered in the various conditions (Fig. EV5K,L). At 28 days after the injury, the size of regenerating myofibers was decreased in animals transplanted with old bone marrow, and this was only partly rescued by young Sepp1$^{\Delta Mac}$ bone marrow (Fig. 5G). The overall collagen I area was not altered in the various conditions (Fig. EV5M), while lipid deposition was decreased in the Sepp1$^{KO}$ condition, but this accounted for less than 3% of the muscle section area (Fig. EV5N). Altogether, these data show that the alteration of the regenerative capacities of the old muscle is dependent of Sepp1 in macrophages for the first steps of the repair process, although additional pathways are likely involved in the immune cell defect observed in the old organism.

## Discussion

Skeletal muscle function is an important determinant in aging. Indeed, exercise or physical activity are recognized now as strategies to improve or to maintain a healthy condition. The capacities of tissues to repair after an injury decline with aging, and we show that skeletal muscle regeneration is impaired in aged mice, as it was previously observed (Sousa-Victor et al, 2022). We show here, as it was observed in other models, that old regenerating muscles are composed of smaller fibers, thus accounting for a decreased muscle mass (Kanazawa et al, 2022; Markworth et al, 2021; Patsalos et al, 2018; Rahman et al, 2020; Sadeh, 1988; Sloboda et al, 2018). This phenotype was particularly robust since it was still observed one month post injury, a time point considered as a full recovery of myofiber size (Varga et al, 2016b). Pioneer studies have evidenced a dysregulation of the myogenic regulating factors MyoD and Myogenin in the old regenerating muscle (Marsh et al, 1997), that was confirmed by the identification of several intracellular signaling pathways whose regulation is impaired in old MuSCs (Bernet et al, 2014; Cosgrove et al, 2014; García-Prat et al, 2016; García-Prat et al, 2020; Price et al, 2014; Sousa-Victor et al, 2014; Tierney et al, 2014). However, skeletal muscle regeneration also relies on the coordinated interactions between MuSCs and their close environment (Hong et al, 2022; Panci and Chazaud, 2021; Singh and Chazaud, 2021). Here, we show that the kinetics of FAPs, ECs and macrophages are altered in terms of number of cells as well as gene expression in the old regenerating muscle. This indicates a general impairment of cell–cell interactions and alteration in the MuSC niche. However, kinetics of differential gene expression in old versus young cells showed specific temporality depending on the cell type. These results highlight the variations of response to age in different cell types, in accordance with large *omics* studies showing that different cells age according to different trajectories and temporality, increasing the complexity in deciphering aging mechanisms at the molecular level (Rutledge et al, 2022).

Changes in gene expression in FAPs were mainly observed at early stages of the regeneration process. Old FAPS expressed lower levels of genes related to the cell cycle and genes associated with stimuli response at steady state and day 2 post injury, suggesting a lower or slower response upon muscle injury. This may be related to the entry into senescent of a subsets of FAPs upon muscle injury (Moiseeva et al, 2023). Consistently, we have previously shown that old FAPs are less proliferative than young FAPs (Lukjanenko et al, 2019). The increased number of FAPs observed in old regenerating muscle at day 7 may be a consequence of this delayed response of the cells to injury in the old muscle. We have also shown that old FAPs aged do not support MuSC myogenesis as young FAPs do, notably because they do not secrete enough matricellular WISP1 (Lukjanenko et al, 2019). FAP-derived fibroblasts are the major source of ECM in the muscle. In vitro, old FAPs are more prone to form fibroblasts than adipocytes (Lukjanenko et al, 2019) and secrete high levels of collagen IV and Laminin (Zwetsloot et al, 2012), components of the basal lamina. We do not find here an increase in Collagen I area in the old regenerating muscle, in accordance with previous reports (Cui et al, 2019), but on the contrary to others (Rahman et al, 2020), likely due to the assessment technique. Nevertheless, our gene expression results show that all cell types decrease their expression of ECM components in the old regenerating muscle. This may make an important alteration in the organization of the ECM itself, rather than of its abundance, as well as changes in ECM-derived cues, some positive signals being lost in aging (Lukjanenko et al, 2019; Schüler et al, 2021). On the contrary to FAPs, old ECs show the highest differential expression of genes during the late steps of regeneration (days 4 and 7 post injury). The decreased number of ECs observed at day 7 may account for the reduced expression of genes of the cell cycle we observed at day 4 post injury. Accordingly, the capillary to fiber ratio was decreased in the old muscle after an acute exercise in humans, as well as the distance with MuSCs (Nederveen et al, 2016), which was shown to be important in muscle homeostasis maintenance (Christov et al, 2007; Latroche et al, 2017; Verma et al, 2018).

Previous studies showed that the number of macrophages is higher in the old regenerating muscle as compared with the young/adult, and inflammatory markers have been also found higher expressed to the detriment of markers of repair macrophages (Markworth et al, 2021; Patsalos et al, 2018; Rahman et al, 2020; Reidy et al, 2019; Sloboda et al, 2018). Our kinetics show that from day 2 post injury, at the time when the resolution of inflammation starts (Varga et al, 2016a), the number of Ly6C$^{pos}$ inflammatory macrophages is higher in the old muscle and still stays higher at day 7 post injury, when the resolution is largely ended. Thus, macrophages have a more pro-inflammatory phenotype and are more numerous in the old regenerating muscles. Our gene expression analysis shows an increase in the expression of

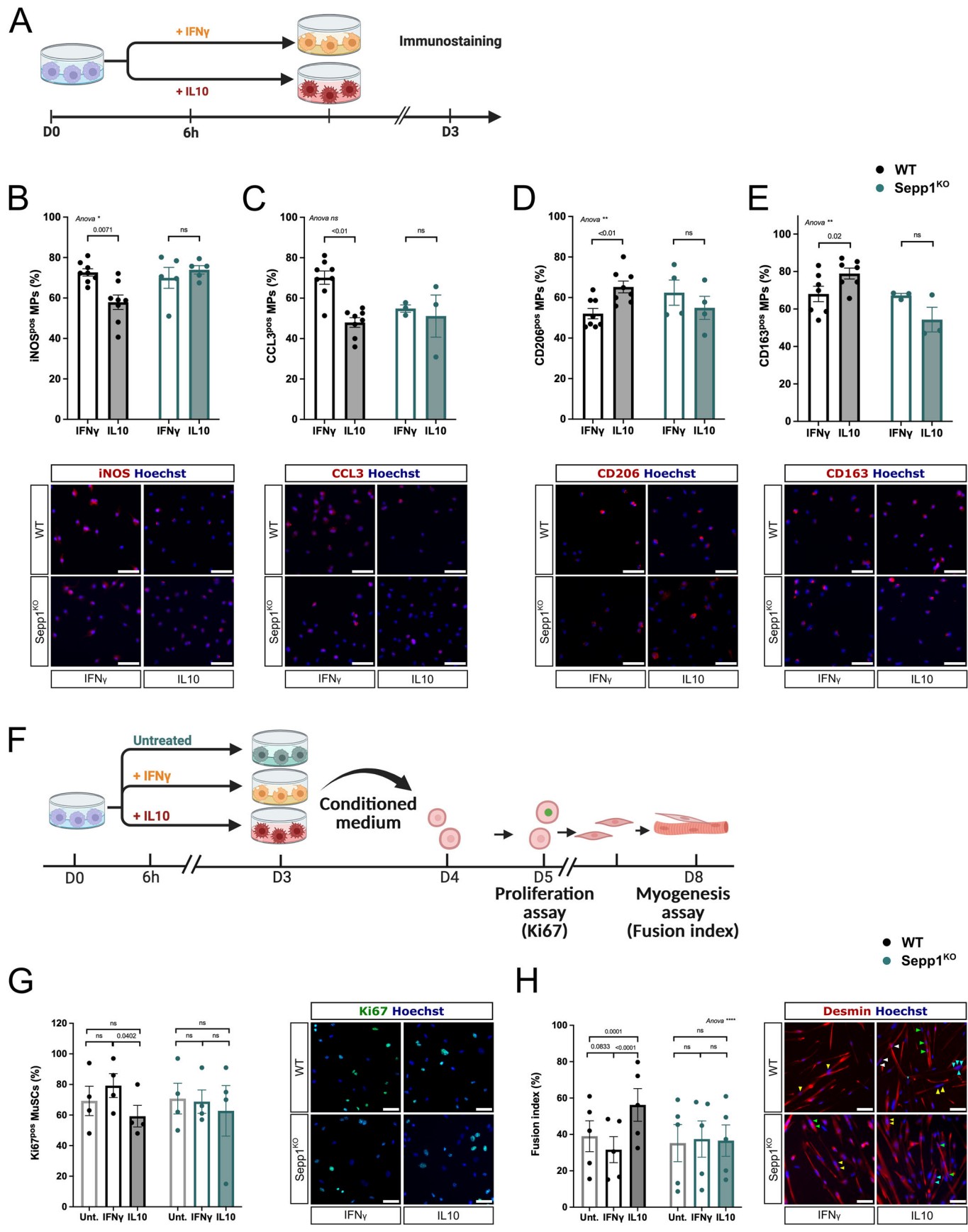

**Figure 4.  Effect of the loss of Sepp1 on macrophage phenotype and functions in vitro.**

(A–E) Wild-type (WT) or Sepp1^KO bone marrow-derived macrophages (BMDMs) were polarized into pro-inflammatory and anti-inflammatory macrophages with IFNγ and IL10, respectively, and were analyzed for their inflammatory status by immunofluorescence. The number of cells expressing the pro-inflammatory markers iNOS (**B**) ($n = 5$–8), CCL3 (**C**) ($n = 3$–8) and the anti-inflammatory markers CD206 (**D**) ($n = 4$–8) and CD163 (**E**) ($n = 3$–7) was counted. A two-way ANOVA test was performed, followed by multiple comparisons using Šidák test. Bars = 40 µm. (**F–H**) WT or Sepp1^KO BMDMs were polarized as in (**A**) or left untreated (Unt.), and conditioned medium was collected and transferred onto Muscle Stem Cells (MuSCs) to evaluate their proliferation (**G**) ($n = 4$) and their myogenesis (**H**) ($n = 5$). A two-way ANOVA test was performed, followed by multiple comparisons using Šidák test. (**H**) Arrowheads show myonuclei within myotubes (one color for one myotube). Bars = 40 µm. Data information: values are given as mean ± SEM. Each dot represents one BMDM culture derived from one mouse. The result of the ANOVA test is shown for each graph as *$P < 0.05$; **$P > 0.01$; ****$P < 0.0001$. Source data are available online for this figure.

inflammatory genes in Ly6C^pos at the early steps of regeneration in the old muscle. Then, Ly6C^neg cells increase the expression of cell cycle genes in the old regenerating muscle, but remain less numerous than in the young animal. Interestingly, pathway enrichment analysis also indicates that Ly6C^neg macrophages present an alteration of their metabolic regulation, which may be of importance in their function as resolving macrophages. Indeed, we and others have previously shown the importance of metabolic regulation in the acquisition of a full functional resolving phenotype by Ly6C^neg macrophages (Giannakis et al, 2019; Juban and Chazaud, 2017; Mounier et al, 2013; Varga et al, 2016b). Two consequences of the alteration of the kinetics of macrophages in the old regenerating muscle are: (i) sustained necrosis. We show here that the old muscle exhibits necrosis at later stages of regeneration, until at least day 7. In other systems, old macrophages present altered phagocytic capacities (De Maeyer and Chambers, 2021), delaying tissue repair and increasing the inflammatory burden; (ii) a general inflammatory context in old regenerating muscle, leading to the expression of an inflammatory signature by all non-immune cell types (FAPs, ECs and MuSCs). This is in accordance with the increased expression of inflammatory genes that was previously observed in the regenerating old muscle tissue (*Cd86*, *Cd80*, *Ccl2*, *IL1b*, *Cxcl10*, *iNOS*, *TNFα*) (Patsalos et al, 2018; Sloboda et al, 2018). Moreover, fibroblasts isolated from old resting rat muscle express inflammatory markers (Zwetsloot et al, 2012).

The failure to resolve inflammation in old muscle likely relies on a variety of causes. A recent study identified mesencephalic astrocyte-derived neurotrophic factor (MANF) as required for the resolution of inflammation and is impaired in old macrophages (Sousa et al, 2023). In other tissues, deficiency in efferocytosis and altered metabolism, which both control the resolution, are observed in old macrophages (Ferrara et al, 2022; Hu et al, 2023; Ryu et al, 2022; Seegren et al, 2023; Sousa et al, 2023). As such, we identified that old macrophages also failed to increase the expression of an antioxidant protein, Selenoprotein P (*Sepp1*) at the time of the resolution of inflammation. Sepp1 is a selenium supplier to cells (Saito and Takahashi, 2002) and is a plasma selenoprotein, which marks selenium levels in the blood and which is able to bind cell membranes (Burk et al, 2003). Selenium deficiency increases oxidative stress and increases inflammatory marker expression (iNOS, IL-1β, IL-12, IL10, PTGE, and NF-κB) and reduces the synthesis of antioxidant enzymes (CAT, T-AOC, SOD, and GSH-Px) (Prabhu et al, 2002; Xu et al, 2020; Zamamiri-Davis et al, 2002) in macrophages in vitro. Selenium deficiency also reduces their phagocytic activity (Xu et al, 2020). On the opposite, adding sodium selenite to macrophages at low concentrations (high concentrations being toxic) increases glutathione peroxidase activity and decreases the SP1 transcription factor activity (Shilo

et al, 2005). It also decreases IKKB and COX2 expression via 1dPGJ2, which triggers PPARγ activity (Vunta et al, 2007), which we have previously shown to be required for the acquisition of a fully functional repair phenotype of macrophages in regenerating muscle (Varga et al, 2016b). Using a mouse model of deficiency for selenocystein tRNA (that is required for the expression of selenoproteins) in myeloid cells (LysMCre;SecKO), it was shown that selenoproteins are required for the resolution of inflammation in the zymosan-induced peritonitis model (Korwar et al, 2021). In accordance with the above supposed functions of Sepp1, we show here that Sepp1 is required for macrophage acquisition of both phenotype and function of the resolving phenotype. Sepp1 is expressed and secreted by numerous cell types but its functions in tissue homeostasis are still elusive, as well as its mechanisms of action. Sepp1 can enter cells through a receptor-mediated uptake mechanism, that was shown to depend on members of the lipoprotein receptor family in testis and kidney (Burk and Hill, 2009). As such, Sepp1 delivers Se to cells and acts as a Se transporter, via its C-terminal moiety, which contains ten selenocysteins (Saito et al, 2004). The N-terminal portion of the molecule exerts an intracellular antioxidant function in the cells (Saito et al, 2004). Our results showed that the full-length Sepp1 protein is required for the phenotypic transition of macrophages since BMDMs mutated for both the antioxidant and the transport functions did not acquire the resolving phenotype and function. In vivo, *Sepp1* deficiency in macrophages leads to a failure of the resolution of inflammation and a delay in muscle regeneration, similar to what is observed in the old regenerating muscle. The regenerated myofibers in LysMCre;Sepp1^KO animals showed lower myonuclei content but similar size than the WT, suggesting additional mechanisms at work in restorative macrophages for delivering their regenerating cues. Moreover, using bone marrow transplantation, we showed that young Sepp1-deficient macrophages are not able to rescue the deficit in muscle regeneration observed in old animals, on the contrary to young WT macrophages that improved this process. The impaired impact on macrophages on MuSCs that we established in vitro was not fully recapitulated in vivo in terms of the number of MuSCs in the regenerating muscle, although the outcome on myofiber regeneration was shown to be partly dependent on macrophagic Sepp1 (maturation of myofibers and size of regenerated myofibers). A dietary Se supplementation of aged mice sounds like an interesting approach but would lack specificity. For instance, Se supplementation stimulates the production of all selenoproteins, not only Sepp1 (Burk and Hill, 2009), including Selenoprotein N that is involved in myogenic differentiation, mitochondrial maintenance and sarcomere organization, which are all altered in the old muscle myofibers (Zito and Ferreiro, 2021).

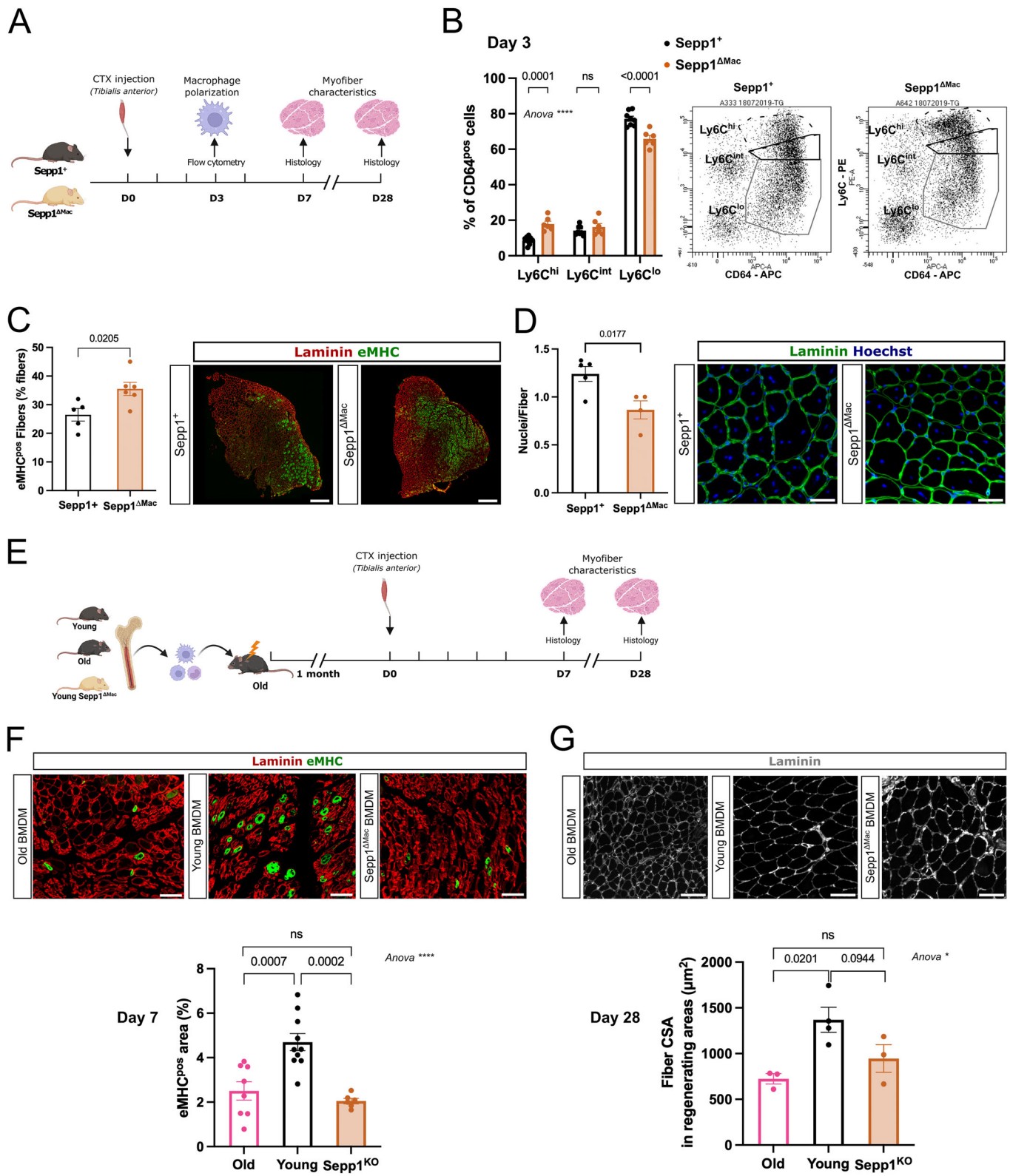

**Figure 5. Effect of the loss of Sepp1 in macrophages on skeletal muscle regeneration in vivo.**

(A–D) *Tibialis Anterior* (TA) muscles from Wild-type (WT) and Sepp1$^{\Delta Mac}$ mice were injected with cardiotoxin and were harvested 3, 7, and 28 days after the injury. (B) The number of Ly6C$^{pos}$, Ly6C$^{int}$ and Ly6C$^{neg}$ macrophages was quantified by flow cytometry at day 3 as a percentage of total CD64$^{pos}$ macrophages ($n = 6$–8). Representative dot plots are shown. A two-way ANOVA test was performed, followed by multiple comparisons using Šidák test. (C) The number of fibers expressing the embryonic myosin heavy chain (eMHC) was counted at day 7 after the injury, as a percentage of the total number of myofibers *per* muscle section ($n = 5$–6). Student $T$ test was performed. Bars = 500 μm. (D) The number of myonuclei present inside myofibers was counted after laminin staining at day 28 after the injury ($n = 4$–5). Student $T$ test was performed. Bars = 40 μm. (E–G) Old WT mice were irradiated and bone marrow transplanted with bone marrow from either young, old or Sepp1$^{\Delta Mac}$ mice and TA muscles were injected with cardiotoxin one month later and were harvested 7 and 28 days after the injury. (F) The area of fibers expressing eMHC was evaluated at day 7 as a percentage of the total damaged/regenerating area ($n = 6$–10). A one-way ANOVA test was performed, followed by multiple comparisons using the Tukey test. Bars = 80 μm. The middle panel is a section of the image shown in Fig. EV5J. (G) The area of myofibers present in regenerating areas was evaluated at day 28. One-way ANOVA test was performed, followed by multiple comparisons using the Tukey test. Bars = 80 μm. Data information: Values are given as mean ± SEM. Each dot represents one TA muscle. The result of the ANOVA test is shown for each graph as *$P < 0.05$; ****$P < 0.0001$. Source data are available online for this figure.

In conclusion, the present study provides a thorough analysis and comparison of gene expression profiles in MuSCs and MuSC niche cells in the regenerating young and old muscles. These analyses uncover the high complexity of aging features in individual cell types, although they share the same tissue environment and highlight the asynchronicity of differential gene expression in the various cell types during tissue repair. The availability of the entire comparative analysis represents a unique tool to decipher the genomic regulation of aging during muscle regeneration in specific cell types. Moreover, the present study uncovers a new function for Sepp1 in macrophages for the resolution of inflammation. Sepp1, which expression is blunted in old macrophages, is required for the acquisition of the phenotype and function of restorative macrophages and the establishment of the regenerative inflammation that is strongly altered in the aged muscle.

## Methods

### Reagents and tools table

| Reagent/resource | Reference or source | Identifier or catalog number |
|---|---|---|
| **Experimental models** | | |
| C57BL/6J | Hill et al, 2003 | |
| Sepp1$^{KO}$ (B6.Cg-Selenoptm1Rfb) | Kurokawa et al, 2014 | |
| Sepp1$^{U40S/U40S}$ (B6.Cg-Selenoptm3.1Rfb) | Hill et al, 2007 | |
| Sepp1$^{\Delta 240-361}$ (B6.Cg-Selenoptm4.1Rfb) | Hill et al, 2012 | |
| Sepp1$^{fl/fl}$ (B6.Cg-Selenoptm3.1Rfb) | | |
| LysM$^{Cre}$ mice (B6.129P2-Lyz2$^{tm1(cre)Ifo}$/J) | | |
| LysM$^{Cre/+}$;Sepp1$^{fl/fl}$ | | |
| CX3CR1$^{gfp/+}$ | | |
| **Antibodies** | | |
| Anti-a7-integrin Alexa Fluor 647-conjugated | AB lab, University of British Columbia | AB0000538 |
| Anti-CCL3 | Santa Cruz | sc-1383 |
| Anti-CD163 | Santa Cruz | sc-33560 |
| Anti-CD206 | Santa Cruz | sc-58987 |
| Anti-CD31 | Abcam | ab7388 |
| Anti-CD31 PE-conjugated | eBioscience | 12-0311-82 |
| Anti-CD34 FITC-conjugated | eBioscience | 11-0341-82 |

| Reagent/resource | Reference or source | Identifier or catalog number |
|---|---|---|
| Anti-CD45 | eBioscience | 25-0451-81 |
| Anti-CD45 PE-Cy7-conjugated | eBioscience | 25-0451-82 |
| Anti-CD64 | BD PharMingen | 558539 |
| Anti-Collagen I | Southern Biotech | 1310-01 |
| Anti-desmin antibody | Abcam | 32362 |
| Anti-eMHC/MYH3 | Santa | sc-53091 |
| Anti-F4/80 | Abcam | ab6640 |
| Anti-iNOS | Abcam | ab3523 |
| Anti-Ki67 | Abcam | 15580 |
| Anti-laminin | Sigma-Aldrich | L9393 |
| Anti-Ly6C | eBioscience | 12-5932-82 |
| Anti-Ly6C antibody APC-conjugated | eBioscience | 17-5932-82 |
| Anti-Pax7 (hybridoma) | DSHB | |
| Anti-PDGFRα | RD systems | AF1062 |
| Anti-perilipin | Abcam | ab3526 |
| Anti-Sca-1 PerCP-Cy5.5- conjugated | eBioscience | 45-5981-82 |
| Cy™3 AffiniPure® Donkey Anti-Mouse IgG (H + L) | Jackson ImmunoResearch | 715-165-150 |
| Alexa Fluor® 488 AffiniPure® Donkey Anti-Mouse IgG (H + L) | Jackson ImmunoResearch | 715-545-150 |
| Alexa Fluor® 488 AffiniPure® Goat Anti-Rabbit IgG (H + L) | Jackson ImmunoResearch | 111-545-003 |
| Cy™3 AffiniPure® Goat Anti-Rabbit IgG (H + L) | Jackson ImmunoResearch | 111-165-003 |
| **Chemicals, enzymes, and other reagents** | | |
| Cardiotoxin CTX | Latoxan | |
| Collagenase B | Roche Diagnostics GmbH | 11088807001 |
| Dispase | Roche Diagnostics GmBH | 0494207800 |
| FcR Blocking reagent | Miltenyi Biotec | 130059901 |
| Fluoromount G mounting medium | Interchim | FP-483331 |
| G/Ultroser | PALL Life Sciences | #15950017 |
| Hoechst solution | Sigma-Aldrich | B2261 |
| IFNγ | RD systems | 485-MI-100 |

| Reagent/resource | Reference or source | Identifier or catalog number |
|---|---|---|
| IL10 | RD systems | 417-ML-005 |
| Matrigel | Corning | 356231 |
| Ovation SoLo RNA-Seq Library Preparation Kit | Nugen/Tecan | |
| Picogreen | Thermofisher | P7589 |
| **Software** | | |
| FIJI | https://imagej.net/ | |
| Bcl2FastQ Demux Pipeline (UBP) (software version v2.19.1). | Illumina | |
| STAR version 2.6.1d defaults. | Dobin et al, 2013 | |
| *featureCounts* version 1.6.4. | Liao et al, 2014 | |
| StringTie version 2.0. | Pertea et al, 2015 | |
| BioMart package. | Durinck et al, 2009 | |
| ESeq2 | Love et al, 2014 | |
| *ComplexHeatmap package version 2.10.0 make_comb_mat* function | Gu et al, 2016 | |
| *PantaRhei* package *version 0.1.2 sankey diagram* function | R | |
| fgsea package version 1.16.0 | Korotkevich et al, 2021 | |

## Mice

Young (10 weeks) and old (24 months) C57BL/6 J males were purchased from Janvier Labs, France. Sepp1$^{KO}$ (B6.Cg-Sele-noptm1Rfb) (Hill et al, 2003); Sepp1$^{U40S/U40S}$ (B6.Cg-Sele-noptm3.1Rfb) (Kurokawa et al, 2014); Sepp1$^{\Delta240-361}$ (B6.Cg-Selenoptm4.1Rfb) (Hill et al, 2007) and Sepp1$^{fl/fl}$ (B6.Cg-Sele-noptm3.1Rfb) (Hill et al, 2012) were kindly provided by Pr Raymond Burk (Vanderbilt University, USA). Sepp1$^{fl/fl}$ mice were crossed with LysM$^{Cre}$ mice (B6.129P2-Lyz2$^{tm1(cre)Ifo}$/J) to make LysM$^{Cre/+}$;Sepp1$^{fl/fl}$ (Sepp1$^{\Delta Mac}$), where Sepp1 is specifically deleted in myeloid cells (controls are LysM$^{+/+}$;Sepp1$^{fl/fl}$ littermates). Mutant mice were used at 8–10 weeks of age, and only males were used for in vivo muscle regeneration experiment. Mice were housed in an environment-controlled facility (12–12 h light–dark cycle, 25 °C), received water and food ad libitum. All the experiments and procedures were conducted in accordance with French and European legislations on animal experimentation and approved by the Local Ethics Committee CEEA-55 and the French Ministry of Agriculture (APAFIS #10463-2017062617107339 & APAFIS #39580-2022102110038352).

## Muscle injury model

Mice were anesthetized in an induction chamber using 4% isoflurane. The hindlimbs were shaved before injection of 50 µl cardiotoxin CTX, (Latoxan, 12 µM) in each Tibialis Anterior (TA) muscle. Mice were euthanized at various time points after the induction of injury.

Bone marrow transplantation was performed as previously described (Mounier et al, 2013). Total bone marrow cells were isolated by flushing of the tibiae and femurs of young (2–3 month-old CX3CR1$^{gfp/+}$ or LysM$^{cre}$;Sepp1$^{fl/fl}$) or old (22–25 month-old C57BL/6) donor males with RPMI 1640/10% FBS. CX3CR1$^{gfp/+}$ mice were used as young WT bone marrow donors to allow engraftment efficiency assessment. Bone marrow cells were transplanted into old (22–25 month-old) C57BL/6 recipient males previously irradiated by gamma rays with a dose of 8.5 Gy on a Synergy apparatus (Elekta). Total bone marrow cells were intravenously injected (10$^7$ cells diluted in 100 µl of RPMI 1640/ 50% mouse serum) in recipient mice as previously described (Mounier et al, 2013). Muscle injury was induced as described above 5 weeks after the transplantation, and TA muscles were harvested 7 and 28 days later. On the day of sacrifice, BMDMs from CX3CR1$^{gfp/+}$ and LysM$^{cre}$;Sepp1$^{fl/fl}$ transplanted animals were generated as described above and used to determine engraftment efficiency by PCR.

## Histology

Muscles were harvested, frozen in liquid nitrogen-precooled isopentane and stored at −80 °C. Ten micrometer-thick cryosections were prepared and treated for (1) hematoxylin-eosin, that was used to discard muscles which have less than 80% of damaged myofibers and (2) immunofluorescence, where the following antibodies were used: anti-laminin (L9393, Sigma-Aldrich), anti-Collagen I (1310-01, Southern Biotech), anti-PDGFRα (AF1062, RD systems), anti-CD31 (ab7388, Abcam), anti-F4/80 (ab6640, Abcam), anti-eMHC/MYH3 (sc-53091, Santa Cruz), anti-Pax7 (hybridoma from DSHB), anti-perilipin (ab3526, Abcam), revealed by secondary antibody conjugated with FITC or Cy3 (Jackson ImmunoResearch) and anti-mouse IgGs conjugated with Cy3 (715-165-150, Jackson ImmunoResearch). Sections were incubated in 1:1000 Hoechst solution (B2261, Sigma-Aldrich) and washed once with PBS before mounting with Fluoromount G mounting medium (FP-483331, Interchim). For the analysis of myofibers, the slides were automatically scanned using a microscope (Axio Observer.Z1, Zeiss) connected to a camera (CCD CoolSNAPHQ2) using Metavue software. The entire muscle section was automatically reconstituted by the Metavue software. Analysis of the myofiber cross-sectional area (CSA) was performed using the Open-CSAM ImageJ macro (Desgeorges et al, 2019) and results are given as mean of the CSA for all myofibers and as the distribution of myofibers according to their area (as a % of all myofibers). For the analysis of mononucleated cells and of collagen deposition, about 15 pictures were randomly taken in the whole section and positive cells harboring a nucleus were counted manually with ImageJ, while collagen area was measured using an ImageJ macro (Juban et al, 2018).

## Flow cytometry

The analysis of myeloid cells was performed as previously described (Juban et al 2018). Briefly, muscles were minced and digested with collagenase B (11088807001, Roche Diagnostics GmbH), the cell suspension was passed through a 30-µm cell strainer and was incubated with FcR Blocking reagent (130059901, Miltenyi Biotec) for 20 min at 4 °C in PBS containing 2% fetal bovine serum (FBS) (10270, Gibco). Cells were then labeled with CD45 (25-0451-81, eBioscience), CD64 (558539, BD PharMingen) and Ly6C (12-5932-

82, eBioscience) antibodies (or isotypic controls) for 30 min at 4 °C before analysis was run using a FACSCanto II flow cytometer (BD Biosciences).

## FACS isolation of cells

MuSCs, FAPs, ECs, neutrophils, Ly6C[pos] macrophages and Ly6C[neg] macrophages were isolated from regenerating muscle as previously described (Juban et al, 2018; Latroche et al, 2018). Briefly, TA muscles were dissociated and digested in DMEM F/12 medium containing 10 mg/ml of collagenase B and 2.4 U/ml Dispase II (#0494207800, Roche Diagnostics GmBH) at 37 °C for 30 min and passed through a 30-µm cell strainer. CD45[pos] and CD45[neg] were separated using magnetic beads. CD45pos cells were incubated with anti-mouse FcgRII/III (2.4G2) and further stained with PE-Cy7-conjugated anti-CD45 (25-0451-82, eBioscience) and APC-conjugated anti-Ly6C antibody (17-5932-82, eBioscience). CD45[neg] cells were stained with PE-Cy7-conjugated anti-CD45, PerCP-Cy5.5- conjugated anti-Sca-1 (45-5981-82, eBioscience), Alexa Fluor 647-conjugated anti-a7-integrin (AB0000538, AB lab, University British Columbia), PE-conjugated anti-CD31 (12-0311-82, eBioscience) and FITC-conjugated anti-CD34 (11-0341-82, eBioscience) antibodies. Cells were sorted using a FACS Aria II cell sorter (BD Biosciences). Flow cytometry plots and gating strategy are available in (Juban et al, 2018; Latroche et al, 2018).

## Bulk mRNA isolation and sequencing library preparation

RNA was extracted from sorted cells. The low amount of available material imposes the usage of a low-input library preparation kit. The samples were subdivided into six different sorted cell types (MuSCs, FAPs, ECs, neutrophils, Ly6C[pos] macrophages and Ly6C[neg] macrophages, four time points (D0, D2, D4, D7) and 2 age conditions (cell isolated from young or old mice). Each of these data point was run in triplicate three different mice. The entire experiment was run in 4 batches of 30 samples and 2 controls (same mix of four cell populations added to each batch of library preparation). Sequencing libraries were prepared using the Ovation SoLo from NuGen. The Ovation SoLo RNA-Seq system integrates NuGen's Insert-Dependent Adaptor Cleavage (InDA-C) technology to provide targeted depletion of unwanted transcripts (rRNA) by specific and robust enzymatic steps. The system also includes an 8 pb barcode for multiplexing followed by an 8 bp randomer for identification of unique molecules (UMI) to remove PCR duplicates from the transcript counting analysis. Libraries are quantified with Picogreen (Life Technologies) and size pattern is controlled with the DNA High Sensitivity Reagent kit on a LabChip GX (Perkin Elmer). Libraries are pooled at an equimolar ratio (i.e. an equal quantity of each sample library) and clustered at a concentration of 20 pmol on paired-end sequencing flow cell CBU2FANXX (Illumina). Sequencing is performed for $2 \times 125$ cycles on a HiSeq 2500 (Illumina) using the SBS V4 chemistry (Sequencing by Synthesis). Primary data quality control is performed during the sequencing run to ensure the optimal flow cell loading (cluster density) and to check the quality metrics of the sequencing run. Sequencing data were demultiplexed and FastQ files and a fastQC report were generated by the Bcl2FastQ Demux Pipeline (UBP) (software version v2.19.1).

## RNAseq analysis—expression quantification

Data preprocessing was carried out using the nf-core pipeline *rnaseq* version 1.4.2 [*nf-core/rnaseq*]. The mouse reference genome used was GRCm38 (mm10), with the corresponding gtf file for exons junctions. Mapping was performed with STAR (Dobin et al, 2013) version 2.6.1 d defaults. Raw counts.txt files were generated via *featureCounts* (Liao et al, 2014) version 1.6.4. Coverage and transcript abundances were estimated via StringTie (Pertea et al, 2015) version 2.0. Cell types and their respective sampling timing are listed as follows (cell type (abbreviation) [Days available]): endothelial (ECs) [D0, D2, D4, D7], fibro-adipogenic precursors (FAPs) [D0, D2, D4, D7], muscle stem cells (MuSCs) [D0, D2, D4, D7], neutrophils (Neutrophils) [D2], inflammatory macrophages (Inflammatory-Mac) [D2, D4] and resolving macrophages (Resolving-Mac) [D2, D4, D7]. Each sample name was defined by a unique combination age.cellType.day (for example: Old.Neutrophils.D2), comprising three biological replicates. Data visualization and analysis were performed using custom Rstudio scripts.

## RNAseq analysis—filtering and normalization step

Gene biotype classification was performed with the BioMart (Durinck et al, 2009) package. Only protein-coding genes were retained. Genes expressed in at least three samples with a raw count greater than five were kept. A hierarchical clustering was done using $\rho$ Spearman's correlation coefficients among TPM (Transcripts per Million) normalized samples. Variance stabilizing transformation (vst) (provided by DESeq2 (Love et al, 2014)) was calculated on raw count matrices at three levels: whole dataset matrix, specific cell-type matrices, and cell-type and time point specific matrices. Principal Component Analysis (PCA) was applied at each level to reveal global effects across libraries.

## RNAseq analysis—differential expression and advanced visualization

Using raw counts, DESeq2 (Love et al, 2014) package was chosen to test differential expression. Firstly, simple contrasts Old vs Young were carried out by day and by cell type, for all cell types and respective available time points (for example, Neutrophils Old vs. Young, on day D2). Significantly differentially expressed genes (DEG) for the Old vs Young contrast were selected by fixing a Benjamini–Hochberg corrected p-value threshold of 0.05 (Padj <= 0.05). Moreover, to assess the timeline flow of significant DEG in each cell type, days summing up one or more significantly differentially expressed genes were extracted (for each gene and each cell type), and their combinations were calculated with *ComplexHeatmap package version 2.10.0 make_comb_mat* (Gu et al, 2016) function. A visual representation was done with the *PantaRhei* package *version 0.1.2 sankey diagram function*. The sense of DEG (up or downregulated) was distinctively represented. Secondly, a differential expression analysis dynamic was proceeded. the contrasts $((t + 1)\_old \text{ vs } (t)\_old)$ vs $((t + 1)\_young \text{ vs } (t)\_young)$ were evaluated inside each cell type, except for neutrophils, as they had been sampled at a single time point.

## RNAseq analysis—gene set enrichment analysis and advanced visualization

Pathway enrichment was performed via fgsea package (Korotkevich et al, 2021) version 1.16.0 using all genes in the whole expression matrix (all cell types gathered), sorted by padj*(log2FoldChange/abs(log2FoldChange) for the statistic option. For the database calling (pathways option), the Molecular Signatures Database (*MSigDB*) was accessed through *msigdbr* package (species Mus musculus, category C2 and subcategory CP:REACTOME). The Reactome pathways list was sorted in hierarchical levels (with ReactomePathwaysRelation.txt and ReactomePathways.txt in https://reactome.org/download-data). This hierarchy was used to represent significantly enriched pathways (*P*adj < 0.05). The combination of day and cell type whose pathways were significantly enriched was calculated.

### BMDM cultures

Bone marrow-derived macrophages (BMDMs) were prepared from WT, Sepp1$^{KO}$, Sepp1$^{U40S/U40S}$ or Sepp1$^{D240-361}$ mice. Polarization/activation of BMDM was performed as described in (Mounier et al, 2013). Briefly, BMDMs were treated with IFNγ 1 μg/mL (485-MI-100, RD systems) or IL10 2 μg/m (417-ML-005, RD systems) for 3 days. BMDMs were fixed and permeabilized before being incubated with the following antibodies: anti-iNOS (ab3523, Abcam), anti-CCL3 (sc-1383, Santa Cruz), anti-CD206 (sc-58987, Santa Cruz) and anti-CD163 (sc-33560, Santa Cruz), revealed by Cy3-conjugated secondary antibodies. Cells were stained with Hoechst and mounted in Fluoromount. About 12-15 pictures were taken randomly, and positive cells were counted using ImageJ software.

### BMDM/muscle stem cell cocultures

BMDMs were obtained as above, and after polarization for 3 days, cells were washed, and a serum-free DMEM medium was added for 24 h to obtain macrophage-conditioned medium (Mounier et al, 2013). MuSCs were obtained from TA muscle as previously described (Theret et al, 2017) and cultured using standard conditions in DMEM/F12 medium containing 20% heat-inactivated FBS and 2% G/Ultroser (Pall Inc.). For proliferation assay, MuSCs were seeded at 10,000 cells/cm$^2$ on a Matrigel coating (1:10) and were incubated for 24 h with macrophage-conditioned medium containing 2.5% FBS. Then cells were immunostained for Ki67 (anti-Ki67 antibody 15580, Abcam) visualized by a Cy3-conjugated secondary antibody. The number of positive cells was counted using ImageJ software. For myogenesis assay, MuSCs were seeded at 30,000 cells/cm$^2$ on Matrigel coating (1:10) and incubated for 3 days with macrophage-conditioned medium containing 2% horse serum. Then, cells were labeled with an anti-desmin antibody (32362, Abcam) visualized by a Cy3-conjugated secondary antibody. The fusion index was calculated as the number of nuclei inside multinucleated cells on the total number of nuclei, using ImageJ software.

### Statistical analysis of experimental procedures

At least three independent experiments in vitro and eight animals in vivo were used, and statistical significance was determined using Student's *t* test and ANOVA. All mice were randomly allocated to groups, and analysis was performed blind to experimental conditions.

## Data availability

The datasets are available at the Gene Expression Omnibus database GSE271744.

The source data of this paper are collected in the following database record: biostudies:S-SCDT-10_1038-S44319-025-00516-3.

## Peer review information

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

## Acknowledgements

The authors thank William Jarassier for help with curating the RNAseq data and training JG. The authors thanks the support from AFM-Telethon (MyoNeurALP Alliance), Fondation pour la Recherche Médicale (DHH and GJ), Agence Nationale de la Recherche (JB), EU EJPRD Myocity (FLG, PM, and JG).

## Author contributions

**Dieu-Huong Hoang**: Formal analysis; Investigation; Visualization; Methodology; Writing—original draft; Writing—review and editing. **Jessica Bouvière**: Formal analysis; Investigation; Methodology; Writing—original draft; Writing—review and editing. **Johanna Galvis**: Data curation; Software; Formal analysis; Investigation; Methodology; Writing—original draft; Writing—review and editing. **Pauline Moullé**: Data curation; Software; Formal analysis; Investigation; Visualization; Methodology; Writing—original draft. **Orane Mercier**: Investigation; Methodology. **Eugenia Migliavacca**: Software; Formal analysis; Investigation; Methodology. **Ananga Ghosh**: Investigation; Methodology. **Gaëtan Juban**: Investigation; Methodology; Writing—review and editing. **Sophie Liot**: Investigation; Methodology. **Pascal Stuelsatz**: Formal analysis; Methodology; Writing—review and editing. **Fabien Le Grand**: Funding acquisition. **Jérôme N Feige**: Conceptualization; Resources; Formal analysis; Supervision; Funding acquisition; Validation; Writing—original draft; Project administration; Writing—review and editing. **Rémi Mounier**: Conceptualization; Resources; Formal analysis; Supervision; Funding acquisition; Validation;

Writing—original draft; Project administration; Writing—review and editing.
**Bénédicte Chazaud**: Conceptualization; Resources; Formal analysis;
Supervision; Funding acquisition; Validation; Visualization; Writing—original
draft; Project administration; Writing—review and editing.

    Source data underlying figure panels in this paper may have individual
authorship assigned. Where available, figure panel/source data authorship is
listed in the following database record: biostudies:S-SCDT-10_1038-S44319-
025-00516-3.

## Disclosure and competing interests statement

DHH, JB, JG, PM, AG, OM, GJ, SL, RM, FLG, and BC declare no competing
interests. EM, PS, and JNF are employees of Société des Produits Nestlé SA.

# Expanded View Figures

**Figure EV1.  Histological analysis of regenerating young and old muscle.**

*Tibialis Anterior* muscles from young (10 weeks old) and old (24 months old) mice were injected or not with cardiotoxin and were harvested 2, 4, 7 and 28 days after injury. (**A**) Mouse body weight was quantified ($n = 3$–9). Two-way ANOVA test was non significant. Multiple unpaired *t* tests were performed and the *P* values are given for each day. (**B**–**I**) The muscle sections were immunostained for various proteins. From laminin immunostaining, the cross-section myofiber area distribution at day 28 after injury (**B**) ($n = 6$), and the total muscle area (**C**) ($n = 4$–6) were measured. Two-way ANOVA test was performed followed by multiple comparisons using Šidák test. (**B**) Multiple unpaired *t* tests were additionally performed and the *P* values are shown in blue. Representative pictures of immunostainings for Laminin (**C**) (bars = 80 μm), IgGs (**E**) (bars = 50 μm), PDGFRα (**F**) (bars = 50 μm), Collagen I (**G**) (bars = 40 μm), CD31 (**H**) (bars = 40 μm), and F4/80 (**I**) (bars = 40 μm). Data information: Values are given as mean ± SEM. Each dot represents one mouse. Result of the two-way ANOVA test is shown for each graph as \*$P < 0.05$; \*\*\*\*$P < 0.0001$.

▶

  

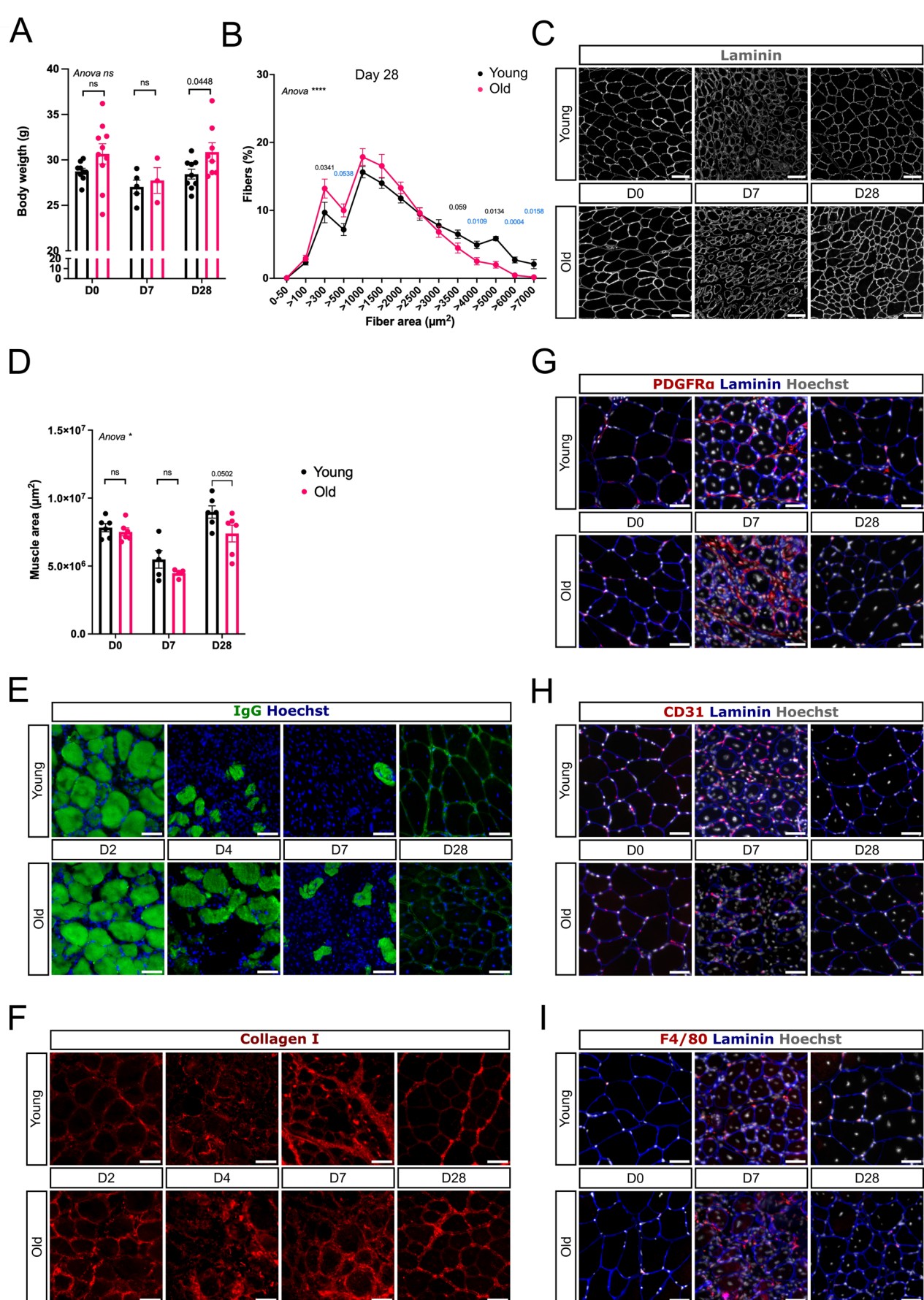

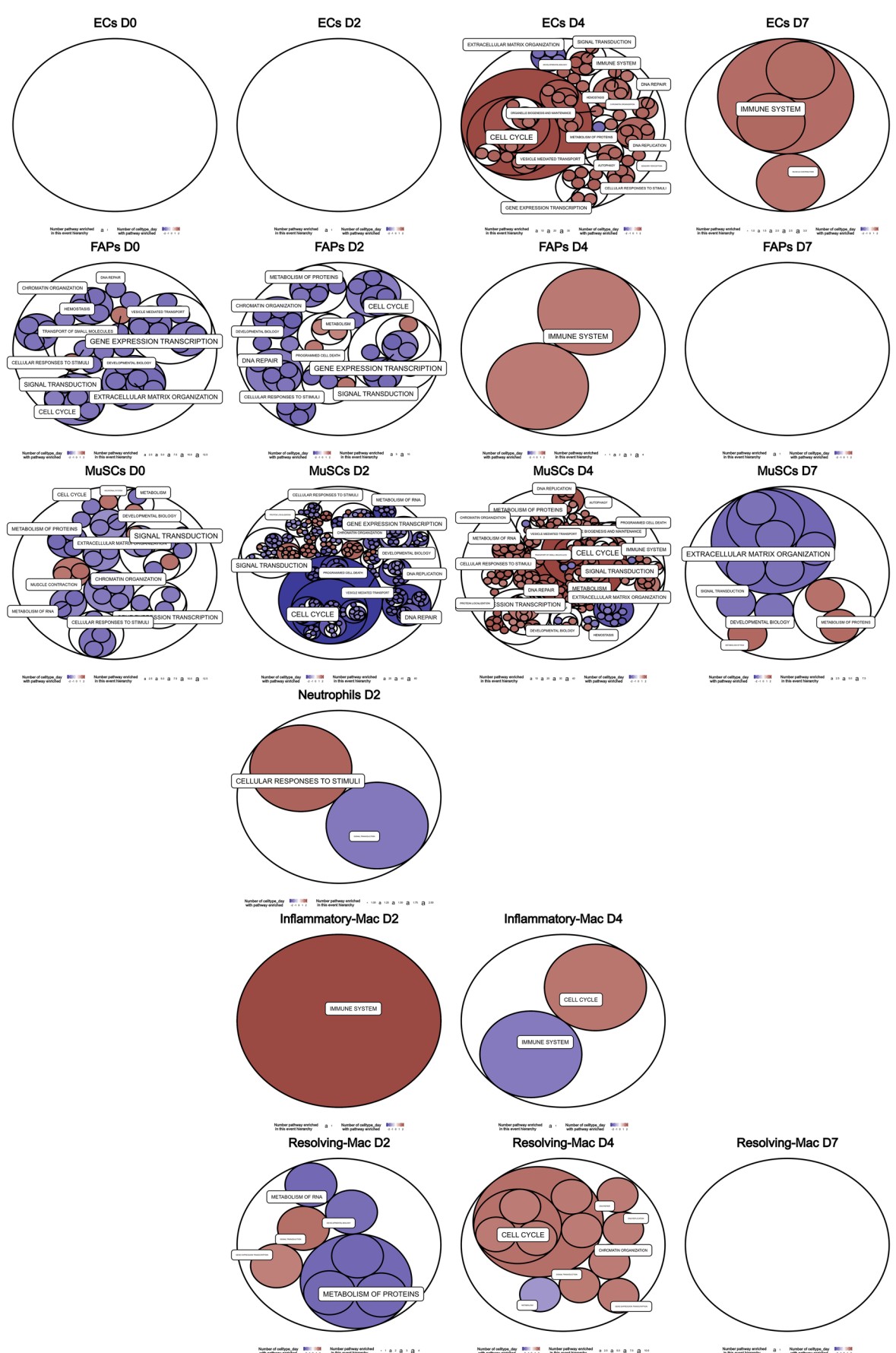

◀ **Figure EV2. Enriched signaling pathways in old *versus* young mononucleated cells.**

Hierarchical overview of Reactome pathway is presented, pathway labels correspond to 25 headers of the hierarchical levels, and the size is scaled based on the number of enriched pathways found in their respective sons. Each circle corresponds to a pathway and its color is the NES.

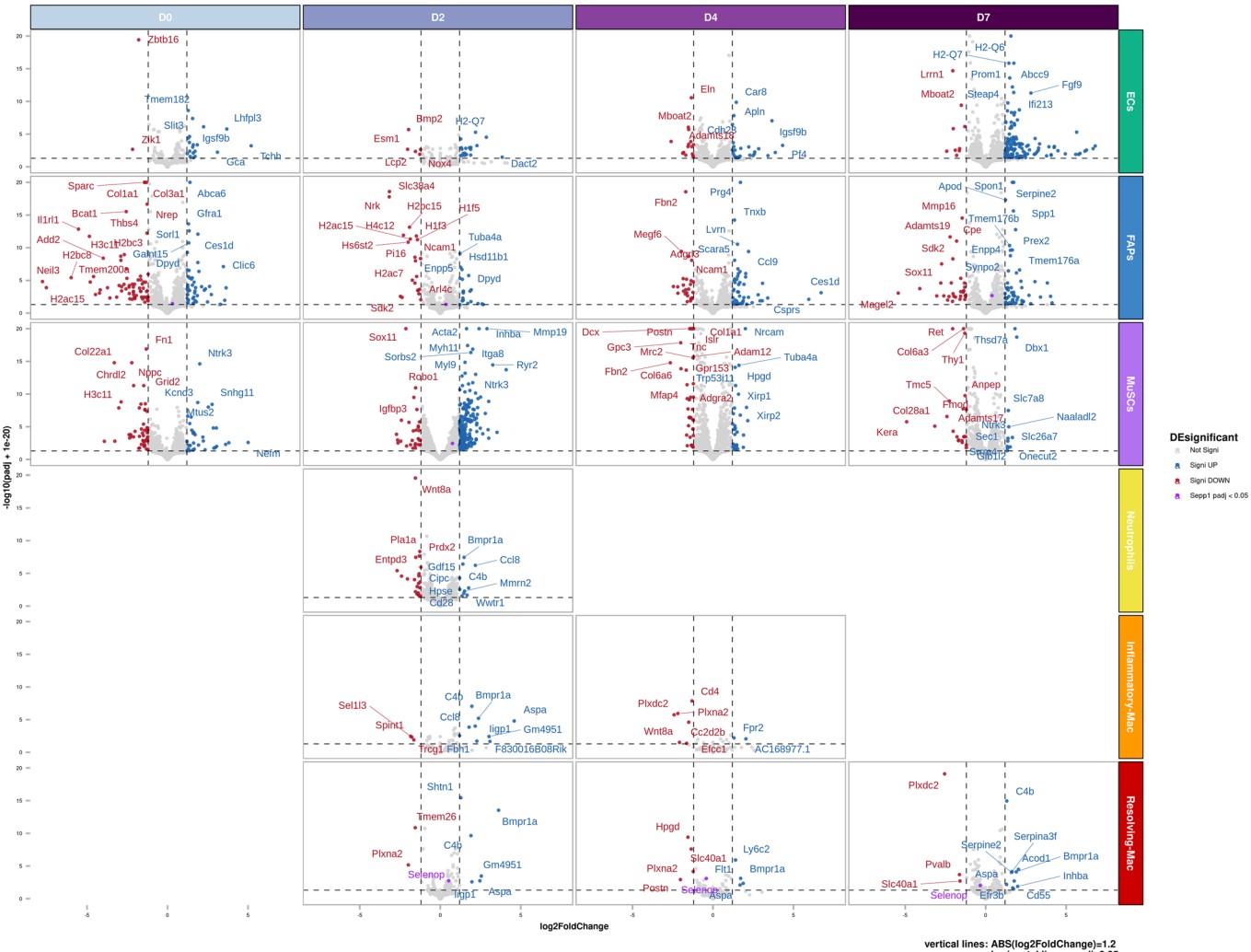

**Figure EV3. Differentially expressed genes (DEG) in old *versus* young mononucleated cells.**

Volcano plot showing log2 fold change (RNAseq) for old *versus* young samples plotted against the –log10 adjusted *P* value (FDR = 0.05) as determined by DESeq2. Significantly differentially expressed genes (DEG) for the Old vs Young contrast were selected by fixing a Benjamini–Hochberg corrected p-value threshold of 0.05 (padj <= 0.05) (*n* = 3).

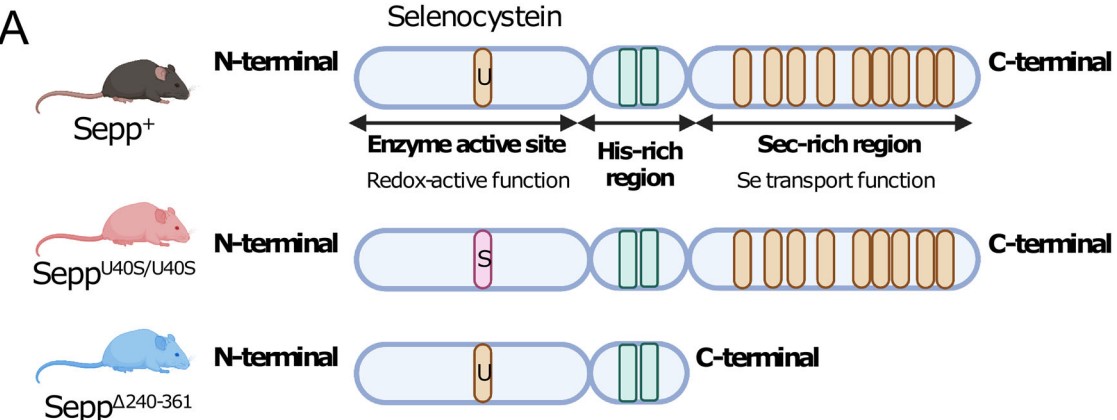

## Macrophage polarization

## Myogenesis

**Figure EV4.  Effect of the loss of redox-activity and selenium transport in Sepp1 on macrophage phenotype and functions in vitro.**

(A) Schematic representing Sepp1 structure and the mouse models having mutation impairing either the redox function (Sepp$^{U40S/U40S}$), or the selenium transport function (Sepp$^{\Delta240-361}$). (B–E) Wild-type (WT), Sepp$^{U40S/U40S}$ and Sepp$^{\Delta240-361}$ bone marrow-derived macrophages (BMDMs) were polarized into pro-inflammatory and anti-inflammatory macrophages with IFNγ and IL10, respectively and analyzed for their inflammatory status by immunofluorescence. The number of Sepp$^{U40S/U40S}$ BMDMs expressing the pro-inflammatory markers iNOS ($n = 6$–8) and CCL3 ($n = 5$–8) (B) and the anti-inflammatory markers CD206 ($n = 7$–8) and CD163 ($n = 4$–7) (C) was counted. The number of Sepp$^{\Delta240-361}$ BMDMs expressing the pro-inflammatory markers iNOS ($n = 4$–8) and CCL3 ($n = 4$–8) (D) and the anti-inflammatory markers CD206 ($n = 5$–8) and CD163 ($n = 4$–7) (E) was counted. (F–I) WT and Sepp$^{U40S/U40S}$ BMDMs were polarized as above and conditioned medium was collected and transferred onto Muscle Stem cells (MuSCs) to evaluate their proliferation (F) ($n = 4$–5) and their myogenesis (G) ($n = 5$–6). (F–I) WT and Sepp$^{\Delta240-361}$ BMDMs were polarized as above and conditioned medium was collected and transferred onto MuSC) to evaluate their proliferation (H) ($n = 4$–5) and their myogenesis (I) ($n = 5$–6). Data information: Values are given as mean ± SEM. Each dot represents one experiment using primary cells issued from one animal. Two-way ANOVA test was performed followed by multiple comparisons using Šidák test. Result of the two-way ANOVA test is shown for each graph as $*P < 0.05$; $**P < 0.01$; $***P < 0.001$; $****P < 0.0001$.

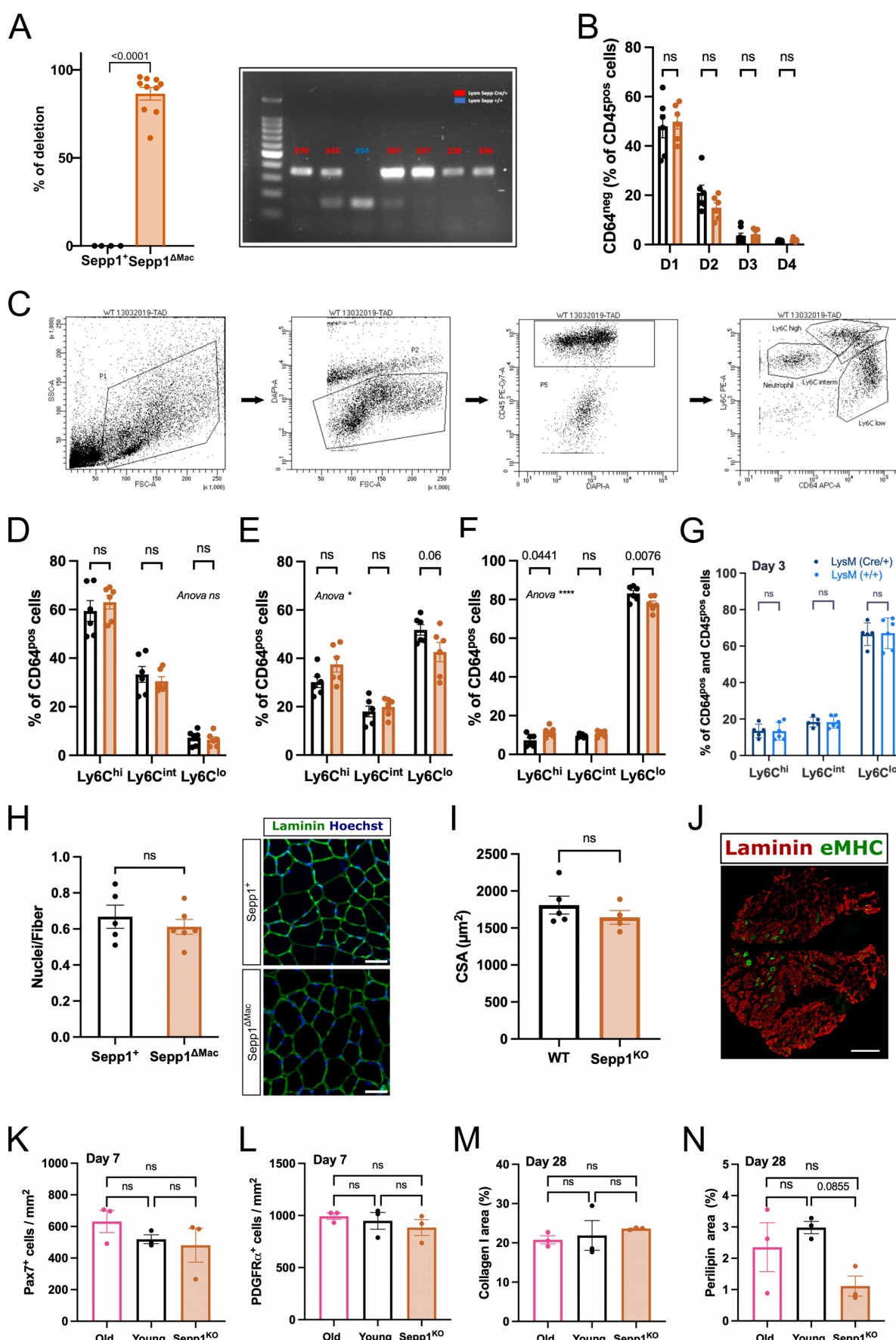

**Figure EV5. Effect of the loss of Sepp1 in macrophages on skeletal muscle regeneration in vivo.**

(A) Evaluation of the depletion of Sepp1 gene in CD11b$^{pos}$ bone marrow cells of Sepp1$^{\Delta Mac}$ mice ($n = 4$–9). Student $t$ test was performed. (B–F) *Tibialis Anterior* (TA) muscles from Wild-type (WT) and Sepp1$^{\Delta Mac}$ mice were injected with cardiotoxin and were harvested 1, 2, 3, 4 days after the injury. (B) The number of CD45$^{pos}$ CD64$^{neg}$ cells (neutrophils) was quantified by flow cytometry as a percentage of total CD45$^{pos}$ immune cells ($n = 6$–8). Two-way ANOVA test was performed followed by multiple comparisons using Šidák test. (C) Gating strategy for the analysis of macrophage subsets by flow cytometry. (D–F) The number of Ly6C$^{pos}$, Ly6C$^{int}$ and Ly6C$^{neg}$ macrophages was quantified by flow cytometry at day 1 (D), 2 (E) and 4 (F) as a percentage of total CD64$^{pos}$ macrophages ($n = 6$). Two-way ANOVA test was performed followed by multiple comparisons using Šidák test. (G) LysM$^{Cre+/+}$ and control (LysM$^{+/+}$) mice were analyzed for the populations of macrophages at day 3 after injury ($n = 5$–6). Two-way ANOVA test was performed followed by multiple comparisons using Šidák test. (H) Uninjured WT and Sepp1$^{\Delta Mac}$ TA muscles were analyzed for the number of nuclei *per* myofiber ($n = 5$–6). Student $T$ test was performed. Bars = 40 μm. (I) Injured WT and Sepp1$^{\Delta Mac}$ TA muscles were analyzed for the size of the regenerating myofibers (CSA) 28 days after injury ($n = 4$–5). Student $T$ test was performed. (J) View of a total muscle section of an old mouse transplanted with young bone marrow, 7 days post injury, embryonic myosin heavy chain (eMHC) is labeled in green (a part of that picture is shown in Fig. 5F, middle panel). Bar = 500 μm. (K–N) Old WT mice were irradiated and bone marrow transplanted with bone marrow from either young, old or Sepp1$^{\Delta Mac}$ mice and TA muscles were injected with cardiotoxin one month later and were harvested 7 and 28 days after the injury ($n = 3$). The number of Pax7$^{pos}$ (K) and PDGFRα (L) was counted at day 7; the area covered by collagen I (M) and perilipin (N) was quantified at day 28 after injury. One-way ANOVA test was performed followed by multiple comparisons using Tukey test. Data information: Values are given as mean ± SEM. Each dot represents one mouse. Result of the two-way ANOVA test is shown for each graph as *$P < 0.05$; ****$P < 0.0001$.

