## [Peer Review File · EMBO Reports]

Immune aging impairs muscle regeneration via macrophage-derived anti-oxidant selenoprotein P

Dieu-Huong Hoang, Jessica Bouvière, Johanna Galvis, Pauline Moullé, Orane Mercier, Eugenia Migliavacca, Ananga Ghosh, Gaetan Juban, Sophie Liot, Pascal Stuelsatz, Fabien Le Grand, Jerome Feige, Rémi Mounier, and Bénédicte Chazaud

Corresponding author(s): Bénédicte Chazaud (benedicte.chazaud@inserm.fr) , Rémi Mounier (remi.mounier@univ-lyon1.fr)

Review Timeline:

Transfer Date:	6th Jan 25
Editorial Decision:	15th Jan 25
Revision Received:	15th May 25
Editorial Decision:	3rd Jun 25
Revision Received:	11th Jun 25
Accepted:	20th Jun 25

Editor: Achim Breiling

**Transaction Report: This manuscript was transferred to
EMBO reports following peer review at Review Commons.**

**Review
COMMONS**

Revision Plan

Manuscript number: RC-2024-02665

Corresponding author(s): Bénédicte Chazaud

1. General Statements

All 3 reviewers state that the significance of the study is high. We agree with most of the experiments they propose, which results will strengthen the manuscript.

2. Description of the planned revisions

Reviewer#1

2. It would be helpful to know if the MuSC population (e.g. Pax7-positive) actually changes in raw number in the young and aged Sepp1 KO macrophages? It appears that there is little changes in FAP numbers, but the authors mainly focus on the macrophage populations. If the impact of Sepp1 on MuSCs is independent from macrophage polarization that would be an important finding to interpretation.

→ We will perform immunostaining of MuSCs using Pax7 and myogenin antibodies to monitor the impact of macrophages Sepp1 deficiency on myogenesis.

3. Sepp1 has been implicated in macrophage motility (Nelson et al., JBC, 2016). Do the authors have any data that the Sepp1 KO macrophages have impaired motility?

→ We will perform in vitro motility experiment in WT and SEPP1KO macrophages (towards damaged muscle protein extract to mimic the physiological situation, as we previously shown, PMID: 36520372).

Reviewer#2

1. To avoid overlooking other major mechanisms, I suggest the authors conduct more in-depth analyses using Sepp1 cKO mice to explore possible interactions between Sepp1 and MuSCs and FAPs, whose interaction with macrophages is known to be critical in muscle injury repair and regeneration.

Specifically, I recommend the following:

a. Examining changes in the expression of MuSC activation and differentiation markers, such as PAX7, MYF5, MYOD, MYOG proteins in injured muscle at different time points, in addition to eMHC, in Sepp1 cKO mice would be informative.

b. Analyzing expression changes of ECM proteins, such as collagens and fat, in Sepp1 cKO muscle using some simple staining techniques.

→ We will perform immunostaining of MuSCs (Pax7 and myogenin directed antibodies) to monitor myogenesis, and FAPs (PDGFRa directed antibodies), collagen (collagen 1 directed antibodies) and lipid droplets (perilipin directed antibodies) to monitor ECM remodeling and fat/adipogenesis.

Reviewer#3

1. The use of wt (or Sepp1 floxed) mice as controls for the Sepp1DMac mice might be inappropriate. Indeed, a possible toxic effect of Cre recombinase expression in macrophages

cannot be excluded, as has been reported in other cre-expressing lines. This might represent a confounding element and assign to Sepp1 ablation a detrimental role that is dependent on cre toxicity. LysMCre mice might be better controls than wt or Sepp1 flox mice and should, in my opinion, be used at least in the key experiments. A similar discussion can also be made for the Sepp1DMac bone marrow transplantations.

→ *We will perform a series of in vivo experiments using the LysM^{Cre/+} mouse (already available in our laboratory) to analyze the time course of macrophage phenotype shifting, as performed in Fig.5B to address the question of Cre toxicity raised by the reviewer.*

3. Description of the revisions that have already been incorporated in the transferred manuscript

Reviewer#1

1. Figure 1C and Supplementary Figure 1B. Please clarify the quantification of the fiber sizes and if they were quantified as a means per fiber cross-sectional area and/or number of fibers? It is unclear the Methodology.

→ *We clarified the quantification of the quantification of fiber cross-sectional area in the method section (p.27).*

Also, please clarify why in Supplementary Figure 1B lists 0.06 as the p-value? Is this supposed to list significance or non-significance with the asterisk(*) labels?

→ *The asterisks and close to 0.05 p values are indicated for each category of fibers as a post-hoc comparison after Anova analysis. The explanation is given in the figure legend.*

Reviewer#2

1. As shown in Fig 5 and Fig. S6, the phenotypes of Sepp1 cKO muscle are striking. eMHC+ myofibers increased by >30%, and myonuclear number per myofiber decreased by >30% in Sepp1 cKO mice. However, the changes of Ly6C+ and Ly6C- macrophage numbers in Sepp1 cKO is quite minor. The Ly6C+ macrophages already form a small population in both in WT and cKO mice, accounting for <20% of total macrophages at D3, and the large population of Ly6C- macrophages shows small differences, with a decrease of <15% in the cKO muscle. The changes are even smaller in cKO muscle at D4 (Fig. S6F).

→ *To our opinion, a direct correlation between the percentage of macrophages shift and the impact on muscle is hard to establish. In previous works, a low change in percentage of Ly6Cpos/neg cells was shown to have a high impact on skeletal muscle regeneration (PMID: 23931756 ; PMID: 25896247 ; PMID: 32183151). We do believe that alteration of the dynamics of macrophage shifting is crucial for tissue repair impairment, more than a « blockade » in one phenotype, as we previously showed (PMID: 27183604). These elements were included page 19.*

2. Regarding the effect of Sepp1 loss on macrophage phenotype and function: The authors compared the number of iNOS+, CCL3+, C206+ and CD163+ macrophages in WT and Sepp1 KO cells treated with IFN γ and IL10 (Fig. 4). In my opinion, this comparison is not particularly helpful for understanding Sepp1 effect on macrophage polarization. The authors should compare 'IFN γ treated - Non-treated - IL10 treated' BMDMs.

Revision Plan

→ *We have done the experiments and added the results of untreated macrophages in the functional assays, Fig4G,4H that confirmed that the impact on MuSC proliferation was driven by IFN γ -treated macrophages and on MuSC fusion by IL-10-treated macrophages (see page 17, new Fig.4).*

Also, the fusion status is difficult to discern in Fig 4H, right panels. Please add arrows to indicate myotube fusion.

→ *This was done.*

Reviewer#3

2. In Figure 5C, the uneven pattern of distribution of eMyHC-positive fibers in the regenerating muscle is suggestive of incomplete damage caused by cardiotoxin injection. This set of experiments should possibly be repeated using a more homogeneous approach, such as cryoinjury, for example.

→ *Cardiotoxin induces the whole muscle damage when the injection proceeds correctly. We discarded the muscles with less than 80% of injured myofibers, since the regeneration process may be expedited below that level. Nevertheless, even when all myofibers are injured, muscle regeneration process follows a centripetal gradient, from the outer regions to the inner regions (PMID: 14715915) and this is what we observed in Fig.5C, with the center of the regenerating area at the worst initially injured point.*

Moreover, a quantification of the cross-sectional area of the fibers in the regenerating area would help in highlighting the claimed delay in regeneration in Sepp1DMac mice. A similar quantification should also be added at 28 days after injury in Sepp1DMac mice to corroborate the data currently reported in panel 5D and suggesting delayed/unproductive regeneration.

→ *We have recently set up an automatic script in FIJI to measures the area of regenerating eMHC^{pos} myofibers. This will be executed on the samples we already have generated.*

Minor comments:

- Page 2, line 6 and page 3, line 9: "FACS" instead of "FACs".

→ *Edit was made.*

- Page 2, line 10: eliminate "the".

→ *Edit was made.*

- Page 3, line 3: in my opinion, a reference/references on the influence of circulating factors on aging-associated impairment in muscle regeneration should be added.

→ *A reference was added*

- Results section, line 5 "weight" instead of "wight". I suggest careful proofreading to fix the typos present in the text.

→ *Edit was made and the text was scrutinized.*

- Suppl. Fig 1: in panels A and B, it is not indicated which is old and which is young.

→ *To keep the figure light, we put the legend only on the right border. We will ask to the editor according to their publishing rules.*

- At the end of page 4, it is stated that "Ly6Cneg restorative macrophages that were less numerous in the old regenerating muscle (-27, -14 and -6% at days 2, 4, 7, respectively, not shown)". It would be nice to have statistical elaboration on this quantification. Please also consider including a graph.

Revision Plan

→ Because $Ly6C^{pos}$ and $Ly6C^{neg}$ cell quantification comes from the same calculation, we assumed that these are the same results (in mirror) and should not be shown. We provide here for the referee's convenience the graph. We will be happy to present it according to the guidelines of the journal.

- The fact that hetero-chronic bone marrow transplantation is impacting on muscle stem cell is known from previous reports. Proper citations should be included.

→ The references for this were already in the text. However, we added a specification concerning MuSCs (page 19).

- I acknowledge the effort put into explaining the study design with words in the M&M section. Nevertheless, the addition of a scheme would help to understand the study design/numerosity of the samples used for the RNA-seq experiments.

→ A new supplemental S2 is now provided.

4. Description of analyses that authors prefer not to carry out

Reviewer#1

4. It would be helpful to know if the Sepp1 KO macrophages have intrinsic functional deficits. Have the authors considered or evaluated the isolated Sepp1 KO macrophages in response to a Dexamethasone stimulus?

→ We have already provided in the study an intrinsic functional deficit of Sepp1KO in Fig.4G,H where it is shown that IL-10 treated Sepp1KO macrophages do not promote inhibition of MuSC proliferation nor stimulation of their fusion. We did not treat macrophages with dexamethasone, but the results would likely be identical. We have previously shown similar polarization profile of macrophages using dexamethasone and IL-10 in vitro and in vivo (PMID: 23931756 and PMID: 36520372).

Reviewer#2

1.

c. The authors should compare phenotypes, including strength tests, of naturally aging WT and Sepp1 cKO muscle, without injury, to better understand the function of Sepp1 in natural muscle aging.

→ We respectfully disagree with the reviewer. Although this question is of interest, it is out of the scope of the present study, which focuses on skeletal muscle regeneration, while natural aging is related to sarcopenia, a biological process highly different from regeneration.

2. Additionally, adding flow cytometry analysis, at least for CD206 and CD163, in addition to IF staining in Fig 4, would strengthen the manuscript.

Revision Plan

→ *We respectfully disagree with the reviewer. The careful counting of positive cells by IF gives the same information than flow cytometry and can avoid some challenges linked to relative normalization to total number of cells with flow cytometry.*

Reviewer#3

I acknowledge that completing the experiments with old mice in a timely manner could be challenging and require the availability of a relatively large number of old mice. Nevertheless, if sufficient old mice are available, I also suggest the following gain of function experiment:

3. Evaluation of muscle regeneration after transplantations of old mice with old bone marrow with (or without, as controls) overexpression of Sepp1 would corroborate the conclusion of the paper and offer a potential therapeutic relevance.

→ *Although the Reviewer's suggestion sounds appealing, there is a strong hurdle, which is the high resistance of macrophages to transfection, rendering overexpression studies barely achievable, unfortunately.*

Minor comments:

- Longitudinal sections of old regenerating muscle at 28 days would clarify if the more numerous and smaller fibers in transverse sections (see Figure 1D and Fig. S1B) reflect the presence of branched fibers, as noticed in the dystrophic context.

→ *Although this question is of high interest, we believe it is beyond the scope of the present study. Indeed, the origin of myofiber branching is still a matter of debate, and the mechanisms are still elusive. It was shown that normal uninjured muscle contains about 15% of branched myofibers at 20-21 mo but after cardiotoxin injury, as early as 4 months, more than 50% of the regenerating myofibers are branched (to be compared with 95% in mdx at the same age) (PMID: 24855558).*

- In various panels (for example, Fig. S2), the font size should be increased to make the text readable.

→ *Increasing the font size would make the figure unreadable. We provided high resolution figure so the reader can zoom in to read the panel of interest. Moreover, we provide a free access to these elements with the online analysis (https://github.com/LeGrand-Lab/Ageing-impact_in_gene_expression_on_skeletal_muscle_repair).*

Dear Dr. Chazaud,

Thank you for the transfer of your research manuscript from Review Commons to EMBO reports. I now went through your manuscript, the referee reports from Review Commons (attached again below) and your revision plan. The referees have several comments, concerns, and suggestions to improve the manuscript, indicating that a major revision of the manuscript is necessary to allow publication of the study.

Going through your revision plan, it seems that the referee points will be adequately addressed during revision. I thus invite you to revise your manuscript accordingly with the understanding that all concerns must be addressed in the revised manuscript and/or in a final detailed point-by-point response (as indicated in your revision plan).

Acceptance of your manuscript will depend on a positive outcome of another round of review using the same set of referees. It is EMBO reports policy to allow a single round of major revision only and acceptance of the manuscript will therefore depend on the completeness of your responses included in the next, final version of the manuscript.

- 1) a .docx formatted version of the final manuscript text (including legends for main figures, EV figures and tables), but without the figures included. Figure legends should be compiled at the end of the manuscript text.
- 2) individual production quality figure files as .eps, .tif, .jpg (one file per figure), of main figures (up to 8) and EV figures (up to 5). Please upload these as separate, individual files upon re-submission.

- 3) one final .docx formatted letter INCLUDING the reviewers' reports and your detailed point-by-point responses to their comments. As part of the EMBO Press transparent editorial process, the point-by-point response is part of the Review Process File (RPF), which will be published alongside your paper.

- 4) a complete author checklist, which you can download from our author guidelines (<https://www.embopress.org/page/journal/14693178/authorguide>). Please insert page numbers in the checklist to indicate where the requested information can be found in the manuscript. The completed author checklist will also be part of the RPF.

- 5) that primary datasets produced in this study (e.g. RNA-seq, ChIP-seq, structural and array data) are deposited in an

appropriate public database. If no primary datasets have been deposited, please also state this in a dedicated section (e.g. 'No primary datasets have been generated and deposited'), see below.

The accession numbers and database should be listed in a formal "Data Availability" section (placed after Materials & Methods) that follows the model below. This is now mandatory (like the COI statement). Please note that the Data Availability Section is restricted to new primary data that are part of this study. This section is mandatory. As indicated above, if no primary datasets have been deposited, please state this in this section

Data availability

8) Regarding data quantification and statistics, please make sure that the number "n" for how many independent experiments were performed, their nature (biological versus technical replicates), the bars and error bars (e.g. SEM, SD) and the test used to calculate p-values is indicated in the respective figure legends (also for potential EV figures and all those in the final Appendix). Please also check that all the p-values are explained in the legend, and that these fit to those shown in the figure. Please provide statistical testing where applicable. Please avoid the phrase 'independent experiment', but clearly state if these were biological or technical replicates. Please also indicate (e.g. with n.s.) if testing was performed, but the differences are not significant. In case n=2, please show the data as separate datapoints without error bars and statistics. See also: <http://www.embopress.org/page/journal/14693178/authorguide#statisticalanalysis>

9) Please also note our reference format:

10) We updated our journal's competing interests policy in January 2022 and request authors to consider both actual and perceived competing interests. Please review the policy <https://www.embopress.org/competing-interests> and update your competing interests if necessary. Please name this section 'Disclosure and Competing Interests Statement' and put it after the Acknowledgements section.

11) We now use CRediT to specify the contributions of each author in the journal submission system. CRediT replaces the author contribution section. Please use the free text box to provide more detailed descriptions and do NOT provide an author contributions section in the revised manuscript text file. See also guide to authors:

<https://www.embopress.org/page/journal/14693178/authorguide#authorshippinguidelines>

12) Please add scale bars of similar style and thickness to microscopic images, using clearly visible black or white bars (depending on the background). Please place these in the lower right corner of the images themselves. Please do not write on or near the bars in the image but define the size in the respective figure legend.

13) Please make sure that all the funding information is also entered into the online submission system and that it is complete

and similar to the one in the acknowledgement section of the manuscript text file.

14) All Materials and Methods need to be described in the main text using our 'Structured Methods' format, which is required for all research articles. According to this format, the Materials and Methods section should include a Reagents and Tools Table (listing key reagents, experimental models, software and relevant equipment and including their sources and relevant identifiers), uploaded as separate file, followed by a Methods and Protocols section in which we encourage the authors to describe their methods using a step-by-step protocol format with bullet points, to facilitate the adoption of the methodologies across labs. More information on how to adhere to this format as well as downloadable templates (.doc or .xls) for the Reagents and Tools Table can be found in our author guidelines (section 'Structured Methods'):

15) Please order the manuscript sections like this, using these names:

Title page - Abstract - Keywords - Introduction - Results - Discussion - Methods - Data availability section - Acknowledgements - Disclosure and Competing Interests Statement - References - Figure legends - Expanded View Figure legends

I look forward to seeing a revised version of your manuscript when it is ready. Please let me know if you have questions or comments regarding the revision.

Best,

Achim Breiling
Senior editor
EMBO reports

Referee #1:

The manuscript by Hoang et al., centers on the evaluation of the selenoprotein P (Sepp1) and its functional role in the regulation of aging and regeneration in skeletal muscle. RNA-sequencing in young and aged mouse muscle satellite cells (MuSCs) revealed altered activation of Sepp1. A macrophage-specific Sepp1 knockout (Sepp1 KO) mouse revealed both in vitro and in vivo deficiencies in macrophage activation and impaired skeletal muscle regeneration. Transplantation studies of young versus old MuSCs from Sepp1 KO and WT mice revealed the inability of young Sepp1 KO MuSCs to restore muscle regeneration suggesting a role for Sepp1 in maintaining the muscle niche through its expression in macrophages.

****General Comments:****

1. Figure 1C and Supplementary Figure 1B. Please clarify the quantification of the fiber sizes and if they were quantified as a means per fiber cross-sectional area and/or number of fibers? It is unclear the Methodology. Also, please clarify why in Supplementary Figure 1B lists 0.06 as the p-value? Is this supposed to list significance or non-significance with the asterisk(*) labels?
2. It would be helpful to know if the MuSC population (e.g. Pax7-positive) actually changes in raw number in the young and aged Sepp1 KO macrophages? It appears that there is little changes in FAP numbers, but the authors mainly focus on the macrophage populations. If the impact of Sepp1 on MuSCs is independent from macrophage polarization that would be an important finding to interpretation.
3. Sepp1 has been implicated in macrophage motility (Nelson et al., JBC, 2016). Do the authors have any data that the Sepp1 KO macrophages have impaired motility?
4. It would be helpful to know if the Sepp1 KO macrophages have intrinsic functional deficits. Have the authors considered or evaluated the isolated Sepp1 KO macrophages in response to a Dexamethasone stimulus?

****Referees cross-commenting****

I agree with Reviewer 3 as well. Methodological clarifications are warranted for the paper.

****Significance: ****

The manuscript is overall well-written and the experiments are comprehensive in scope. Statistical analysis and power calculations are appropriate for a study of this nature. I have a few questions with regards to ruling out alternative interpretation of some of the findings in the Sepp1 macrophage KO mice. Most to nearly-all of the functional assessments on the Sepp1 KO macrophage mice are comprehensive and well-interpreted. If the authors can address my critiques, I think the manuscript will be an excellent addition to the muscle aging and regeneration fields. Sepp1 function likely impacts the inflammatory responses in both normal muscle regeneration and neuromuscular diseases.

With regards to my expertise, I have evaluated several clinical and pre-clinical models of neuromuscular disease in zebrafish and mice. I have profiled key immune populations in response to neuromuscular disease and drug treatments. As I stated above, I believe this manuscript would be an excellent addition to the muscle regeneration and aging fields in its final form.

Referee #2:

In this manuscript, the authors report that aging alters MuSC niche cells in a cell type-specific manner. Among the various cell types, macrophages show particularly prominent changes during muscle aging. The authors discovered significantly reduced expression of Selenoprotein P (Sepp1) in macrophages in regenerating aged muscle. Macrophage-specific knockout of Sepp1 resulted in an increase in regenerating myofibers but a decrease in cross-sectional area, likely due to impaired acquisition of the repair profile in macrophages. In subsequent bone marrow transplantation experiments, the authors further found that bone marrow from young WT mice, but not from young Sepp1 cKO, restores muscle regeneration to youthful levels in old mice. The findings are novel, and the manuscript provides valuable insights into the function of macrophage-derived Sepp1 in muscle regeneration during aging. However, I have several suggestions:

1. As shown in Fig 5 and Fig. S6, the phenotypes of Sepp1 cKO muscle are striking. eMHC+ myofibers increased by >30%, and myonuclear number per myofiber decreased by >30% in Sepp1 cKO mice. However, the changes of Ly6C+ and Ly6C- macrophage numbers in Sepp1 cKO is quite minor. The Ly6C+ macrophages already form a small population in both in WT and cKO mice, accounting for <20% of total macrophages at D3, and the large population of Ly6C- macrophages shows small differences, with a decrease of <15% in the cKO muscle. The changes are even smaller in cKO muscle at D4 (Fig. S6F).

My main concern is that explaining Sepp1 cKO phenotypes solely as a 'loss of resolution of inflammation' may be overly ambitious and may not address the true underlying mechanism behind the Sepp1 cKO phenotypes.

To avoid overlooking other major mechanisms, I suggest the authors conduct more in-depth analyses using Sepp1 cKO mice to explore possible interactions between Sepp1 and MuSCs and FAPs, whose interaction with macrophages is known to be critical in muscle injury repair and regeneration.

Specifically, I recommend the following:

- a. Examining changes in the expression of MuSC activation and differentiation markers, such as PAX7, MYF5, MYOD, MYOG proteins in injured muscle at different time points, in addition to eMHC, in Sepp1 cKO mice would be informative.
 - b. Analyzing expression changes of ECM proteins, such as collagens and fat, in Sepp1 cKO muscle using some simple staining techniques.
 - c. The authors should compare phenotypes, including strength tests, of naturally aging WT and Sepp1 cKO muscle, without injury, to better understand the function of Sepp1 in natural muscle aging.
2. Regarding the effect of Sepp1 loss on macrophage phenotype and function: The authors compared the number of iNOS+, CCL3+, C206+ and CD163+ macrophages in WT and Sepp1 KO cells treated with IFN γ and IL10 (Fig. 4). In my opinion, this comparison is not particularly helpful for understanding Sepp1 effect on macrophage polarization. The authors should compare 'IFN γ treated - Non-treated - IL10 treated' BMDMs. Additionally, adding flow cytometry analysis, at least for CD206 and CD163, in addition to IF staining in Fig 4, would strengthen the manuscript.

Also, the fusion status is difficult to discern in Fig 4H, right panels. Please add arrows to indicate myotube fusion.

**Significance: **

The findings are novel, and the manuscript provides valuable insights into the function of macrophage-derived Sepp1 in muscle regeneration during aging.

Referee #3:

In this manuscript, Hoang and colleagues investigate in young and old mice the dynamics of gene expression during muscle regeneration in various mononucleated cells participating in the regenerative process, including different subtypes of macrophages. They also extended the study by focusing their attention on Selenoprotein P (Sepp1), which resulted in being differentially expressed in old and young macrophages. By using a macrophage-specific genetic ablation approach, the collected evidence supports the role of this protein in controlling the resolution of inflammation that is impaired in aging muscle. By performing bone marrow transplantation, they demonstrated that in the absence of Sepp1, the young bone marrow cannot improve the regenerative ability of the old muscle. Collectively, these data increase our knowledge of the cellular and molecular events controlling muscle regeneration and are becoming inefficient in aging. While finding the data and conclusions of interest

for the field of scientists studying the mechanisms associated with muscle aging, the second part of the manuscript, which is centered on the role of Sepp1, might be strengthened further. Specifically, I highlight a few weaknesses below and indicate possible experiments that would corroborate the paper's conclusions.

****Major comments:****

1. The use of wt (or Sepp1 floxed) mice as controls for the Sepp1D^{Mac} mice might be inappropriate. Indeed, a possible toxic effect of Cre recombinase expression in macrophages cannot be excluded, as has been reported in other cre-expressing lines. This might represent a confounding element and assign to Sepp1 ablation a detrimental role that is dependent on cre toxicity. LysMCre mice might be better controls than wt or Sepp1 flox mice and should, in my opinion, be used at least in the key experiments. A similar discussion can also be made for the Sepp1D^{Mac} bone marrow transplantations.

2. In Figure 5C, the uneven pattern of distribution of eMyHC-positive fibers in the regenerating muscle is suggestive of incomplete damage caused by cardiotoxin injection. This set of experiments should possibly be repeated using a more homogeneous approach, such as cryoinjury, for example. Moreover, a quantification of the cross-sectional area of the fibers in the regenerating area would help in highlighting the claimed delay in regeneration in Sepp1D^{Mac} mice. A similar quantification should also be added at 28 days after injury in Sepp1D^{Mac} mice to corroborate the data currently reported in panel 5D and suggesting delayed/unproductive regeneration.

I acknowledge that completing the experiments with old mice in a timely manner could be challenging and require the availability of a relatively large number of old mice. Nevertheless, if sufficient old mice are available, I also suggest the following gain of function experiment:

3. Evaluation of muscle regeneration after transplantations of old mice with old bone marrow with (or without, as controls) overexpression of Sepp1 would corroborate the conclusion of the paper and offer a potential therapeutic relevance.

****Minor comments:****

- Pag 2, line 6 and pag3, line 9: "FACS" instead of "FACs".
- Pag 2, line 10: eliminate "the".
- Pag 3, line 3: in my opinion, a reference/references on the influence of circulating factors on aging-associated impairment in muscle regeneration should be added.
- Results section, line 5 "weight" instead of "wight". I suggest careful proofreading to fix the typos present in the text.
- Suppl. Fig 1: in panels A and B, it is not indicated which is old and which is young.
- At the end of page 4, it is stated that "Ly6Cneg restorative macrophages that were less numerous in the old regenerating muscle (-27, -14 and -6% at days 2, 4, 7, respectively, not shown)". It would be nice to have statistical elaboration on this quantification. Please also consider including a graph.
- Longitudinal sections of old regenerating muscle at 28 days would clarify if the more numerous and smaller fibers in transverse sections (see Figure 1D and Fig. S1B) reflect the presence of branched fibers, as noticed in the dystrophic context.
- In various panels (for example, Fig. S2), the font size should be increased to make the text readable.
- The fact that hetero-chronic bone marrow transplantation is impacting on muscle stem cell is known from previous reports. Proper citations should be included.
- I acknowledge the effort put into explaining the study design with words in the M&M section. Nevertheless, the addition of a scheme would help to understand the study design/numerosity of the samples used for the RNA-seq experiments.

****Referees cross-commenting****

I agree with the suggestion of the other Reviewers of performing more in-depth analyses using Sepp1 cKO mice to explore possible interactions between Sepp1 and MuSCs and FAPs.

****Significance: ****

This study offers an analysis of the gene expression dynamics of various mononucleated cells that are known to drive muscle regeneration. Notably, this analysis has been comparatively performed at different time points of injury in old and young murine muscles. I am a cellular biologist with long-lasting expertise in muscle regeneration, and I don't have the expertise to fully evaluate the bioinformatic elaboration of the gene expression data. Nevertheless, I highlight that the cells were isolated through FACS, and bulk RNAseq was performed, ensuring an analysis of poorly represented transcripts that often are lost in single-cell RNA seq. This comprehensive view of the cellular and molecular events governing regeneration will be accessible to the scientific community. It will be a precious tool for scientists studying aging and tissue regeneration.

Particularly novel is the second part of the study, which ascribes to Sepp1 a role in the resolution of inflammation, which is functional to productive muscle regeneration in young mice. This study extends previous studies ascribing to macrophages' subtype-switch a role in controlling muscle regeneration. Moreover, previous literature indicates that cells derived from the young bone marrow can have beneficial effects on muscle stem cells in aging mice. This concept is extended here by ascribing a role in this context to macrophages and to their expression of Sepp1. Although not completely developed, this finding opens to the

design of potential novel molecular interventions to ameliorate muscle regeneration in aging.

This study will not only be of interest to the scientists studying muscle aging and regeneration. Indeed, macrophage dynamics are also crucial for productive regeneration in other tissues and different pathophysiological settings. Therefore, the audience interested in this work comprehends a vast population of cellular biologists studying stem cells and immune cells.

EMBOR-2025-61118V1-T
Responses to the referee's comments

Referee #1:

The manuscript by Hoang et al., centers on the evaluation of the selenoprotein P (Sepp1) and its functional role in the regulation of aging and regeneration in skeletal muscle. RNA-sequencing in young and aged mouse muscle satellite cells (MuSCs) revealed altered activation of Sepp1. A macrophage-specific Sepp1 knockout (Sepp1 KO) mouse revealed both in vitro and in vivo deficiencies in macrophage activation and impaired skeletal muscle regeneration. Transplantation studies of young versus old MuSCs from Sepp1 KO and WT mice revealed the inability of young Sepp1 KO MuSCs to restore muscle regeneration suggesting a role for Sepp1 in maintaining the muscle niche through its expression in macrophages.

****General Comments:****

1. Figure 1C and Supplementary Figure 1B. Please clarify the quantification of the fiber sizes and if they were quantified as a means per fiber cross-sectional area and/or number of fibers? It is unclear the Methodology.

→ *As requested, we clarified the quantification of the quantification of fiber cross-sectional area in the methods and protocols section (p.14).*

Also, please clarify why in Supplementary Figure 1B lists 0.06 as the p-value? Is this supposed to list significance or non-significance with the asterisk(*) labels?

→ *The asterisks and close to 0.05 p values are indicated for each category of fibers as a post-hoc comparison after Anova analysis. The explanation is now given in the figure legends.*

2. It would be helpful to know if the MuSC population (e.g. Pax7-positive) actually changes in raw number in the young and aged Sepp1 KO macrophages? It appears that there is little changes in FAP numbers, but the authors mainly focus on the macrophage populations. If the impact of Sepp1 on MuSCs is independent from macrophage polarization that would be an important finding to interpretation.

→ *As requested, we performed the experiments (Fig.EV5KL) and modified the text accordingly page 20. We observed no difference between the conditions, indicating that other mechanisms are at work, as mentioned in the text page 9 and in the discussion page 12.*

→ *For the referee's convenience, we provide below another result we obtained, that showed that selenium treatment (Sodium selenite 5 µg/l for 3 days) of MuSCs did not impact on their myogenesis. We prefer not to include those data since there were performed on human primary MuSCs and may add confusion to the manuscript. Nevertheless, we believe that these data reinforce the fact that the impact of selenium acts mainly through macrophages in the context of muscle regeneration.*

3. Sepp1 has been implicated in macrophage motility (Nelson et al., JBC, 2016). Do the authors have any data that the Sepp1 KO macrophages have impaired motility?

→ *The above mentioned study refers to an in vivo indirect evaluation of motility (clearance of parasites after infection). As requested, we performed 2 assays of in vitro motility*

experiments of BMDMs (in vivo motility is impossible to address after muscle injury given the high inflammatory response induced by toxin). Using transwell assay and chemotactic assay (Ibidi chemotactic chambers), we were, unfortunately, unable to measure any migratory behavior of BMDMs (whether WT or SEPP1^{KO}). As chemo-attractants, we used both serum and muscle DAMPS (early injured muscle protein extract that induces macrophages into a pro-inflammatory profile, as we previously showed, PMID: 36520372). We concluded that such migration experiments are hardly doable with differentiated macrophages, which are highly adherent cells, on the contrary to monocytes and macrophagic cell lines that are most often used in the literature.

4. It would be helpful to know if the Sepp1 KO macrophages have intrinsic functional deficits. Have the authors considered or evaluated the isolated Sepp1 KO macrophages in response to a Dexamethasone stimulus?

→ We have already provided in the study an intrinsic functional deficit of Sepp1KO macrophages in Fig.4G,H where it is shown that IL-10 treated Sepp1KO macrophages do not promote inhibition of MuSC proliferation nor stimulation of their fusion, whereas this is the case for WT macrophages. We did not treat macrophages with dexamethasone, but the results would likely be identical as we have previously shown similar polarization profile of macrophages using dexamethasone and IL-10 in vitro and in vivo (PMID: 23931756 and PMID: 36520372).

Referee #2:

In this manuscript, the authors report that aging alters MuSC niche cells in a cell type-specific manner. Among the various cell types, macrophages show particularly prominent changes during muscle aging. The authors discovered significantly reduced expression of Selenoprotein P (Sepp1) in macrophages in regenerating aged muscle. Macrophage-specific knockout of Sepp1 resulted in an increase in regenerating myofibers but a decrease in cross-sectional area, likely due to impaired acquisition of the repair profile in macrophages. In subsequent bone marrow transplantation experiments, the authors further found that bone marrow from young WT mice, but not from young Sepp1 cKO, restores muscle regeneration to youthful levels in old mice. The findings are novel, and the manuscript provides valuable insights into the function of macrophage-derived Sepp1 in muscle regeneration during aging. However, I have several suggestions:

1. As shown in Fig 5 and Fig. S6, the phenotypes of Sepp1 cKO muscle are striking. eMHC+ myofibers increased by >30%, and myonuclear number per myofiber decreased by >30% in Sepp1 cKO mice. However, the changes of Ly6C+ and Ly6C- macrophage numbers in Sepp1 cKO is quite minor. The Ly6C+ macrophages already form a small population in both in WT and cKO mice, accounting for <20% of total macrophages at D3, and the large population of Ly6C- macrophages shows small differences, with a decrease of <15% in the cKO muscle. The changes are even smaller in cKO muscle at D4 (Fig. S6F).

→ To our opinion, a direct correlation between the percentage of macrophages shift and the impact on muscle is hard to establish. In previous works, a low change in percentage of Ly6C^{pos/neg} cells was shown to have a high impact on skeletal muscle regeneration (PMID: 23931756 ; PMID: 25896247 ; PMID: 32183151), demonstrating that even a modest alteration of the dynamics of macrophage shifting is crucial for tissue repair impairment (more than a "blockade" in one phenotype, as we previously showed (PMID: 27183604)). This element was included page 8.

My main concern is that explaining Sepp1 cKO phenotypes solely as a 'loss of resolution of inflammation' may be overly ambitious and may not address the true underlying mechanism behind the Sepp1 cKO phenotypes.

To avoid overlooking other major mechanisms, I suggest the authors conduct more in-depth analyses using Sepp1 cKO mice to explore possible interactions between Sepp1 and MuSCs

and FAPs, whose interaction with macrophages is known to be critical in muscle injury repair and regeneration.

Specifically, I recommend the following:

a. Examining changes in the expression of MuSC activation and differentiation markers, such as PAX7, MYF5, MYOD, MYOG proteins in injured muscle at different time points, in addition to eMHC, in Sepp1 cKO mice would be informative.

b. Analyzing expression changes of ECM proteins, such as collagens and fat, in Sepp1 cKO muscle using some simple staining techniques.

→ *We have performed the requested immunostaining experiments that appear in Fig.EV5KLMN. We found no changes of the parameters, except a decrease in perilipin content in the Sepp1^{KO} conditions but with a very low percentage in a context where ectopic fat accumulation is minimal (>2%). We modified the text accordingly in the results (page 9) and in the discussion (page 26).*

c. The authors should compare phenotypes, including strength tests, of naturally aging WT and Sepp1 cKO muscle, without injury, to better understand the function of Sepp1 in natural muscle aging.

→ *The current manuscript focusses on the cross-talk of aging with niche cells during regeneration. Given that Sepp1 is strongly upregulated during regeneration, we would like to respectfully disagree with the reviewer regarding the scope of this question. Although this question is of interest, we believe it would be better addressed in a separate study on natural aging and chronic inflammation independently of regeneration, as sarcopenia is a biological process highly different from regeneration.*

2. Regarding the effect of Sepp1 loss on macrophage phenotype and function: The authors compared the number of iNOS+, CCL3+, C206+ and CD163+ macrophages in WT and Sepp1 KO cells treated with IFN γ and IL10 (Fig. 4). In my opinion, this comparison is not particularly helpful for understanding Sepp1 effect on macrophage polarization. The authors should compare 'IFN γ treated - Non-treated - IL10 treated' BMDMs.

→ *As requested, we added the results (the experiments were previously done) of untreated macrophages in the functional assays in parallel of IFN γ and IL-10 stimulation (Fig.4G,4H). These results confirmed that the impact of Sepp1 loss on MuSC proliferation was driven by IFN γ -treated macrophages and on MuSC fusion by IL-10-treated macrophages (see new Fig.4G,H).*

Additionally, adding flow cytometry analysis, at least for CD206 and CD163, in addition to IF staining in Fig 4, would strengthen the manuscript.

→ *We respectfully disagree with the reviewer. The careful counting of positive cells by IF gives the same information than flow cytometry and can avoid some challenges linked to relative normalization to total number of cells with flow cytometry. Moreover, we provide in Fig.4F-H a functional readout of macrophage phenotype (impact of macrophage conditioned medium on MuSC proliferation or fusion) that confirms the phenotypic characterization done by IF.*

Also, the fusion status is difficult to discern in Fig 4H, right panels. Please add arrows to indicate myotube fusion.

→ *As requested, arrows were added in Fig.4H images to highlight nuclei in fusing myotubes.*

Referee #3:

In this manuscript, Hoang and colleagues investigate in young and old mice the dynamics of gene expression during muscle regeneration in various mononucleated cells participating in the regenerative process, including different subtypes of macrophages. They also extended the study by focusing their attention on Selenoprotein P (Sepp1), which resulted in being differentially expressed in old and young macrophages. By using a macrophage-specific

genetic ablation approach, the collected evidence supports the role of this protein in controlling the resolution of inflammation that is impaired in aging muscle. By performing bone marrow transplantation, they demonstrated that in the absence of Sepp1, the young bone marrow cannot improve the regenerative ability of the old muscle. Collectively, these data increase our knowledge of the cellular and molecular events controlling muscle regeneration and are becoming inefficient in aging. While finding the data and conclusions of interest for the field of scientists studying the mechanisms associated with muscle aging, the second part of the manuscript, which is centered on the role of Sepp1, might be strengthened further. Specifically, I highlight a few weaknesses below and indicate possible experiments that would corroborate the paper's conclusions.

****Major comments:****

1. The use of wt (or Sepp1 floxed) mice as controls for the Sepp1DMac mice might be inappropriate. Indeed, a possible toxic effect of Cre recombinase expression in macrophages cannot be excluded, as has been reported in other cre-expressing lines. This might represent a confounding element and assign to Sepp1 ablation a detrimental role that is dependent on cre toxicity. LysMCre mice might be better controls than wt or Sepp1 flox mice and should, in my opinion, be used at least in the key experiments. A similar discussion can also be made for the Sepp1DMac bone marrow transplantations.

→ As requested, we have performed a series of in vivo experiments using the LysM^{Cre/+} mouse (already available in our laboratory) to analyze the time course of macrophage phenotype shifting, as performed in Fig.5B to address the question of Cre toxicity raised by the reviewer. The results, presented in Fig.EV5G showed no impact of Cre expression on macrophage polarization during the time course of muscle regeneration. The results were added page 8.

2. In Figure 5C, the uneven pattern of distribution of eMHC-positive fibers in the regenerating muscle is suggestive of incomplete damage caused by cardiotoxin injection. This set of experiments should possibly be repeated using a more homogeneous approach, such as cryoinjury, for example.

→ Cardiotoxin (50µl) induces the whole muscle damage when the injection proceeds correctly. To ensure homogeneity of the regeneration process between samples, we discarded the muscles with less than 80% of injured myofibers, since the regeneration process may be expedited below that level. We added this precision in the Methods and Protocols section page 13. Nevertheless, even when all myofibers are injured, muscle regeneration process follows a centripetal gradient, from the outer regions to the inner regions (PMID: 14715915) and this is what we observed in Fig.5C, with the center of the regenerating muscle area being the region where regeneration completes last. Here below for the referee's convenience are two other examples (left WT, right, Sepp1MacKO)

Moreover, a quantification of the cross-sectional area of the fibers in the regenerating area would help in highlighting the claimed delay in regeneration in Sepp1D^{Mac} mice. A similar quantification should also be added at 28 days after injury in Sepp1D^{Mac} mice to corroborate the data currently reported in panel 5D and suggesting delayed/unproductive regeneration.

→ *As requested, we calculated the CSA at day 28 after injury and found only a modest decrease in the size of the myofibers between the two genotypes, indicating a transient impact of Sepp1 deficiency in macrophages on the overall regeneration process. Data are presented in Fig.EV5I and the result section, page 8, and discussed page 12.*

I acknowledge that completing the experiments with old mice in a timely manner could be challenging and require the availability of a relatively large number of old mice. Nevertheless, if sufficient old mice are available, I also suggest the following gain of function experiment:

3. Evaluation of muscle regeneration after transplantations of old mice with old bone marrow with (or without, as controls) overexpression of Sepp1 would corroborate the conclusion of the paper and offer a potential therapeutic relevance.

→ *Although the Referee's suggestion sounds appealing, there is a strong hurdle, which is the high resistance of macrophages to transfection, rendering overexpression studies barely achievable, unfortunately.*

****Minor comments:****

- Pag 2, line 6 and pag3, line 9: "FACS" instead of "FACs".

→ *As requested, edit was made.*

- Pag 2, line 10: eliminate "the".

→ *As requested, edit was made.*

- Pag 3, line 3: in my opinion, a reference/references on the influence of circulating factors on aging-associated impairment in muscle regeneration should be added.

→ *As requested, a reference was added.*

- Results section, line 5 "weight" instead of "wight". I suggest careful proofreading to fix the typos present in the text.

→ *As requested, edit was made and the text was scrutinized.*

- Suppl. Fig 1: in panels A and B, it is not indicated which is old and which is young.

→ *As requested, edit was made.*

- At the end of page 4, it is stated that "Ly6C^{neg} restorative macrophages that were less numerous in the old regenerating muscle (-27, -14 and -6% at days 2, 4, 7, respectively, not shown)". It would be nice to have statistical elaboration on this quantification. Please also consider including a graph.

→ *Because Ly6C^{pos} and Ly6C^{neg} cell quantification comes from the same calculation, we assumed that these are the same results (in mirror) and should not been shown. We provide here for the referee's convenience the graph. We will be happy to present it according to the guidelines of the journal.*

- Longitudinal sections of old regenerating muscle at 28 days would clarify if the more numerous and smaller fibers in transverse sections (see Figure 1D and Fig. S1B) reflect the presence of branched fibers, as noticed in the dystrophic context.

→ Although this question is of high interest, we believe it is beyond the scope of the present study. Indeed, the origin of myofiber branching is still a matter of debate, and the mechanisms are still elusive. It was shown that normal uninjured muscle contains about 15% of branched myofibers at 20-21 mo. But after cardiotoxin injury, as early as 4 months, more than 50% of the regenerating myofibers are branched (to be compared for instance with 95% in mdx at the same age) (PMID: 24855558). In addition, quantifying branching would require an extensive amount of tedious work using serial section analysis given the way our samples were stored.

- In various panels (for example, Fig. S2), the font size should be increased to make the text readable.

→ Increasing the font size would make the figure unreadable. We provided high resolution figure so the reader can zoom in to read the panel of interest. Moreover, we provide pages 5 and 6 a link to access freely to these elements with the online analysis (https://github.com/LeGrand-Lab/Ageing-impact_in_gene_expression_on_skeletal_muscle_repair).

- The fact that hetero-chronic bone marrow transplantation is impacting on muscle stem cell is known from previous reports. Proper citations should be included.

→ The references for this were already in the text. However, we added a specification concerning MuSCs (page 9).

- I acknowledge the effort put into explaining the study design with words in the M&M section. Nevertheless, the addition of a scheme would help to understand the study design/numerosity of the samples used for the RNA-seq experiments.

→ As requested, a scheme detailing the RNA-seq experimental plan is now provided in the Appendix Fig.S1.

Dear Dr. Chazaud,

Thank you for the submission of your revised manuscript to our editorial offices. I have now received the reports from the three referees that were asked to re-evaluate the study, you will find below. As you will see, referees #1 and #3 now support the publication of your manuscript in EMBO reports. In contrast, referee #2 states that major concerns raised were not adequately addressed.

Upon further consultation the referee indicated that s/he refers to the analysis of MuSCs in the Sepp1 cKO mice (at least through IF staining of a few MuSC markers) and flow cytometry using antibodies such as CD206 or CD163. Going through your p-b-p-response, I note that you consider both points as out of scope of the present study. I would agree, and taking into account that the two other referees are satisfied with the revision, I will thus continue with the submission.

Before we can proceed with formal acceptance, I have these editorial requests I ask you to address in a final revised manuscript:

- Please provide the abstract written in present tense throughout.

- We now use CRediT to specify the contributions of each author in the journal submission system. CRediT replaces the author contribution section. Please use the free text box to provide more detailed descriptions and do NOT provide your final manuscript text file with an author contributions section. See also our guide to authors: <https://www.embopress.org/page/journal/14693178/authorguide#authorshipguidelines>

- Please order the manuscript sections like this, using these names:

Title page - Abstract - Keywords - Introduction - Results - Discussion - Methods - Data availability section - Acknowledgements - Disclosure and Competing Interests Statement - References - Figure legends - Expanded View Figure legends

- Per journal policy, we do not allow 'data not shown', which is stated twice in the manuscript (page 4). All data referred to in the paper should be displayed in the main or Expanded View figures, or an Appendix. Thus, please add these data (or change the text accordingly if these data are not central to the study). See:

<https://www.embopress.org/page/journal/14693178/authorguide#unpublisheddata>

- Please check again that the number "n" for how many independent experiments were performed, their nature (biological versus technical replicates), the bars and error bars (e.g. SEM, SD) and the test used to calculate p-values is indicated in the respective figure legends. Please also check that all the p-values are explained in the legend, and that these fit to those shown in the figure. Please provide statistical testing where applicable. Please avoid the phrase 'independent experiment' but clearly state if these were biological or technical replicates. Please also indicate (e.g. with n.s.) if testing was performed, but the differences are not significant. In case n=2, please show the data as separate datapoints without error bars and statistics. See also:

<http://www.embopress.org/page/journal/14693178/authorguide#statisticalanalysis>

If n<5, please show single datapoints for diagrams. Presently several diagrams have only partial statistics and/or miss the 'n.s.'. Please check. Moreover:

- Please note that the exact p values are not provided in the legends of figures 1A-J; 4B-E, G, H; 5B ,C, D, F, G; EV1 A, B; EV4 B, C, D, E, F, G, H, I; EV5 A, F.

- Please indicate the statistical test used for data analysis in the legend of figure EV3.

- Please note that information related to n is missing in the legend of figure EV3.

- Please add scale bars of similar style and thickness to microscopic images (main and EV figures), using clearly visible black or white bars (depending on the background). Please place these in the lower right corner of the images themselves. Please do not write on or near the bars in the image but define the size in the respective figure legend. Presently, the most scale bars are too small/thin or faint. Please check.

- Please remove now the referee token from the data availability section and add a direct link to the dataset GSE271744. Please also make sure the data is public latest upon online publication of the manuscript.

- The middle panel in Fig. 5F seems to be a mirrored and rotated section of panel EV5J. Please check. If this re-use is intentional, please clearly indicate this in the respective figure legends.

- Thanks for providing the requested source data. Please upload this as one folder per figure, grouping together all the files for this figure (and ZIPed together).

In addition, I would need from you uploaded separately:

- a short, two-sentence summary of the manuscript (not more than 35 words).

- two to four short (!) bullet points highlighting the key findings of your study (two lines each).
- a schematic summary figure as separate file that provides a sketch of the major findings (not a data image) in jpeg or tiff format (with the exact width of 550 pixels and a height of not more than 400 pixels) that can be used as a visual synopsis on our website.

Best,

Referee #1:

The authors have made substantial clarifications and provided additional data in support of their overarching hypothesis and findings. I appreciate the authors providing some data with regards to demonstrating no enhanced myogenic differentiation in the Sepp1 KO mice. Although I would have liked to see additional probing of this area, I will concede that this could comprise a separate study. I have no additional recommendations or concerns with this manuscript.

Referee #2:

The authors did not adequately address the major concerns I raised in my earlier comments.

Referee #3:

Overall, the data presented by the authors is convincing. I believe that this paper could represent an interesting addition to the field and is therefore worth being published.

All editorial and formatting issues were resolved by the authors.

Dr. Bénédicte Chazaud
Université Claude Bernard Lyon 1
Institut NeuroMyoGène
8 Avenue Rockefeller
Lyon 69008
France

Dear Dr. Chazaud,

I am very pleased to accept your manuscript for publication in the next available issue of EMBO reports. Thank you for your contribution to our journal.

Yours sincerely,
